# p53-induced RNA-binding protein ZMAT3 inhibits transcription of a hexokinase to suppress mitochondrial respiration in human cancer cells

Ravi Kumar[1], Simon Couly[2], Bruna R Muys[1], Xiao Ling Li[1], Ioannis Grammatikakis[1], Ragini Singh[1], Mary Guest[3], Xinyu Wen[4], Wei Tang[5], Stefan Ambs[5], Lisa M Jenkins[6], Erica C Pehrsson[7], Raj Chari[3], Tsung-Ping Su[2], Ashish Lal[1]*

[1]Regulatory RNAs and Cancer Section, Genetics Branch, Center for Cancer Research (CCR), National Cancer Institute (NCI), National Institutes of Health (NIH), Bethesda, United States; [2]Cellular Pathobiology Section, Integrative Neuroscience Branch, National Institute on Drug Abuse (NIDA), NIH, Baltimore, United States; [3]Genome Modification Core, Frederick National Lab for Cancer Research, NCI, NIH, Frederick, United States; [4]Oncogenomics Section, Genetics Branch, CCR, NCI, NIH, Bethesda, United States; [5]Laboratory of Human Carcinogenesis, CCR, NCI, NIH, Bethesda, United States; [6]Mass Spectrometry Section, Laboratory of Cell Biology, CCR, NCI, NIH, Bethesda, United States; [7]Advanced Biomedical Computational Science, Frederick National Laboratory for Cancer Research, Frederick, United States

*For correspondence:
ashish.lal@nih.gov

## eLife Assessment

In this study, the authors investigate the role of ZMAT3, a p53 target gene, in tumor suppression and RNA splicing regulation. Using quantitative proteomics, the authors uncover that ZMAT3 knockout leads to upregulation of HKDC1, a gene linked to mitochondrial respiration, and that ZMAT3 suppresses HKDC1 expression by inhibiting c-JUN-mediated transcription. This set of **convincing** evidence reveals a **fundamental** mechanism by which ZMAT3 contributes to p53-driven tumor suppression by regulating mitochondrial respiration.

**Abstract** The tumor suppressor p53 is a transcription factor that controls the expression of hundreds of genes. Emerging evidence indicates that the p53-induced RNA-binding protein ZMAT3 acts as a key splicing regulator that contributes to p53-dependent tumor suppression in vitro and in vivo. However, the mechanism by which ZMAT3 functions within the p53 pathway remains largely unclear. Here, we discovered a function of ZMAT3 in inhibiting transcription of *HKDC1*, a hexokinase that regulates glucose metabolism and mitochondrial respiration in human cancer cells. Quantitative proteomics revealed HKDC1 as the most significantly upregulated protein in *ZMAT3*-depleted colorectal cancer cells. *ZMAT3* depletion resulted in increased mitochondrial respiration, which was rescued by simultaneous depletion of *HKDC1*, suggesting that HKDC1 is a critical downstream effector of *ZMAT3*. Unexpectedly, ZMAT3 did not bind to *HKDC1* RNA or DNA; however, proteomic analysis of the ZMAT3 interactome identified its interaction with the oncogenic transcription factor JUN. ZMAT3 depletion enhanced JUN binding to the *HKDC1* locus, leading to increased *HKDC1* transcription that was rescued upon *JUN* depletion, suggesting that JUN activates *HKDC1* transcription in ZMAT3-depleted cells. Collectively, these findings uncover a mechanism by which ZMAT3 regulates transcription through JUN and demonstrate that *HKDC1* is a

key component of the ZMAT3-regulated transcriptome in the context of mitochondrial respiration regulation.

## Introduction

*TP53* is the most frequently mutated gene in human cancer and functions as a major tumor suppressor (*Kandoth et al., 2013*). *TP53* mutations in the germline of Li-Fraumeni patients and in sporadic cancer are mostly missense mutations that occur in the DNA-binding domain, resulting in loss of tumor suppressor function and, in some cases, gain of oncogenic functions (*Olivier et al., 2002*; *Malkin et al., 1990*; *Oren and Prives, 2024*). Deletion of the *Trp53* gene in mice results in spontaneous tumor development within 6 months of age at 100% penetrance, underscoring the importance of the p53 protein as a tumor suppressor (*Donehower et al., 1992*; *Donehower, 1996*). Mechanistically, p53 functions as a sequence-specific transcription factor activating the expression of hundreds of genes that control diverse cellular processes, including but not limited to cell cycle arrest, apoptosis, senescence, and DNA repair (*Boutelle and Attardi, 2021*; *Indeglia and Murphy, 2024*). Despite the undisputed role of p53 in tumor suppression, our understanding of how p53 target genes mediate the effects of p53 is not fully understood.

Among the p53 target genes, some such as *p21* (*CDKN1A*) control p53-dependent cell cycle arrest, whereas *PUMA* and *NOXA* are critical in inducing apoptosis downstream of p53 (*Waldman et al., 1995*; *Yu et al., 2003*; *Shibue et al., 2003*). In vivo studies in mice have demonstrated that *Abca1*, *Gls2*, *Mlh1*, *Padi4*, and *Zmat3* play key roles in mediating the tumor suppressor effects of p53 (*Indeglia et al., 2023*; *Janic et al., 2018*; *Moon et al., 2019*; *Suzuki et al., 2022*). However, triple knockout mice lacking *p21*, *Puma*, and *Noxa*, which are regulators of cell cycle arrest and apoptosis in the p53 pathway, do not develop tumors spontaneously, unlike p53 null mice (*Valente et al., 2013*). This has led to the search for new mechanisms and effectors of p53-mediated tumor suppression.

An emerging potent mediator of p53 is the p53 target gene *ZMAT3*, which functions as an RNA-binding protein, and studies in mice strongly implicate *Zmat3* as a tumor suppressor (*Bieging-Rolett et al., 2020*; *Bieging-Rolett and Attardi, 2021*; *Brennan et al., 2024*). Mechanistically, using transcriptome-wide approaches from crosslinked cells, we and others recently reported that ZMAT3 directly binds to intronic sequences in thousands of pre-mRNAs and regulates alternative splicing (*Bieging-Rolett et al., 2020*; *Muys et al., 2021*). ZMAT3 has also been shown to interact with AU-rich elements in the 3′ untranslated regions (UTR) of target mRNAs either stabilizing its targets or promoting their decay (*Vilborg et al., 2009*; *Bersani et al., 2016*; *Kim et al., 2012*; *Bersani et al., 2014*). A deeper understanding of the molecular mechanisms by which ZMAT3 functions is necessary to better understand how ZMAT3 functions in p53-mediated tumor suppression.

Here, to better understand the function of ZMAT3, we identified the ZMAT3-interactome and the proteins regulated by ZMAT3. Unexpectedly, this approach revealed that ZMAT3 inhibits mitochondrial respiration by interacting with the transcription factor JUN (c-Jun) to inhibit transcription of the hexokinase *HKDC1* (hexokinase domain containing 1). We focused on *HKDC1* because by quantitative proteomics, HKDC1 was the most strongly up-regulated protein in ZMAT3-depleted colorectal cancer (CRC) cells. Hexokinases are the first rate-limiting enzymes in the glucose metabolic pathway that phosphorylate glucose to glucose-6-phosphate and thereby modulate glycolysis, oxidative phosphorylation, and the pentose phosphate pathway. There are four classic hexokinases (HK1-4), and recently HKDC1 was identified as the fifth hexokinase (*Khan et al., 2018*). Besides its hexokinase function, HKDC1 also interacts with the mitochondrial membrane to maintain mitochondrial homeostasis and plays an important role in preventing cellular senescence (*Cui et al., 2024*; *Khan et al., 2022*; *Zapater et al., 2022*). Although HKDC1 is reported to be overexpressed in many cancers and high HKDC1 expression is associated with poor clinical outcome (*Khan et al., 2022*; *Zapater et al., 2022*; *Wang et al., 2020*; *Wang et al., 2023*), how HKDC1 expression is regulated remains largely unclear. Our findings provide novel insights on the regulation and function of HKDC1 in the p53 pathway via transcriptional inhibition by a ZMAT3/JUN axis.

## Results

### The hexokinase HKDC1 is the most strongly upregulated protein in ZMAT3-knockout cells

To investigate the mechanism by which ZMAT3 promotes growth suppression, we utilized CRISPR/Cas9 to deplete ZMAT3 in HCT116 cells (CRC) using two sgRNAs flanking the p53 response element (RE) in the second intron of *ZMAT3*. The rationale for deleting the p53RE was based on previous data showing that p53RE of the *ZMAT3* gene is critical for ZMAT3 expression (*Muys et al., 2021*). Deleting the p53RE should, therefore, result in a marked decrease in ZMAT3 expression without disrupting the entire *ZMAT3* locus. Although this approach did not completely delete the region spanning the two sgRNAs, a ~57 bp region near sgRNA#1 was deleted, resulting in >75% decrease in *ZMAT3* mRNA levels (*Figure 1A and B*). At the protein level, ZMAT3 was strongly induced only in *ZMAT3*-WT (wild-type) cells upon treatment with Nutlin, a small molecule that upregulates p53 and its target genes (*Figure 1—figure supplement 1A*). As expected, upon Nutlin treatment, p53 and its canonical target p21 were induced to similar levels in *ZMAT3*-WT and isogenic *ZMAT3*-KO (knockout) cells (*Figure 1—figure supplement 1A*, *Figure 1—figure supplement 2A–B*, *Figure 1—figure supplement 1—source data 1*, *Figure 1—figure supplement 2—source data 1*). Depletion of ZMAT3 resulted in increased proliferation and clonogenicity, consistent with previous reports (*Figure 1C*, *Figure 1—figure supplement 1B*; *Bieging-Rolett and Attardi, 2021*; *Brennan et al., 2024*; *Muys et al., 2021*).

To identify the genes regulated by ZMAT3, we next performed RNA-seq from biological triplicates of *ZMAT3*-WT and *ZMAT3*-KO HCT116 cells. As expected, *ZMAT3* mRNA levels significantly decreased (~7.5 fold) in *ZMAT3*-KO cells (*Supplementary file 1*); the recently identified ZMAT3 target gene *MDM4* (*Bieging-Rolett et al., 2020*) was modestly but significantly upregulated in *ZMAT3*-KO cells (*Supplementary file 1*). Transcriptome-wide, upon loss of ZMAT3, 606 genes were significantly up-regulated (adj. p<0.05 and 1.5-fold change) and 552 were down-regulated with a median fold change of 1.76 and 0.55 for the up- and down-regulated genes, respectively (*Figure 1D* and *Supplementary file 1*). Because we and others recently reported that ZMAT3 directly regulates alternative splicing (*Bieging-Rolett et al., 2020*; *Muys et al., 2021*), we reasoned that ZMAT3-dependent changes in splicing could lead to altered protein levels without altering mRNA levels. We, therefore, performed global quantitative proteomics from *ZMAT3*-WT and *ZMAT3*-KO HCT116 cells. At the protein level, 228 proteins were significantly (p<0.05) up-regulated and 108 were down-regulated upon loss of ZMAT3 (*Figure 1E*; *Supplementary file 2*).

ZMAT3 directly binds to intronic sequences in pre-mRNAs and can inhibit inclusion of the neighboring exon (*Bieging-Rolett et al., 2020*; *Muys et al., 2021*). Depletion of ZMAT3 can, therefore, result in increased target gene expression. Gene set enrichment analysis (GSEA) for the proteins upregulated in *ZMAT3*-KO cells revealed glycolysis as the most significantly overrepresented biological process (*Figure 1F*). GSEA analysis from our RNA-seq data showed that glycolysis was among the top 10 overrepresented biological processes for the mRNAs upregulated in *ZMAT3*-KO cells (*Figure 1—figure supplement 1C*). Interestingly, the most strongly upregulated protein in *ZMAT3*-KO cells was the hexokinase HKDC1 (~3.4-fold, *p*<0.05) (*Figure 1E and G* and *Supplementary file 2*). The increase in *HKDC1* expression was also observed at the mRNA level; *ZMAT3* mRNA downregulation in *ZMAT3*-KO cells served as positive control (*Figure 1H*, *Figure 1—figure supplement 1D* and *Supplementary file 1*). We, therefore, chose to focus on *HKDC1* because it was the most strongly upregulated protein upon loss of ZMAT3 and is directly involved in regulating glucose metabolism and mitochondrial respiration that are cellular processes not previously associated with ZMAT3.

### Inhibition of *HKDC1* expression by ZMAT3 is conserved and observed across diverse cell types

To validate inhibition of *HKDC1* expression by ZMAT3, we next performed RT-qPCR from *ZMAT3*-WT and *ZMAT3*-KO HCT116 cells. We observed a ~ 4-fold upregulation of *HKDC1* mRNA upon *ZMAT3* depletion (*Figure 2A*). This upregulation was further confirmed at the protein level by immunoblotting (*Figure 2B*, *Figure 2—source data 1*). Because these experiments were conducted from a single *ZMAT3*-KO clone, we next analyzed our recently published RNA-seq data (*Muys et al., 2021*) conducted in biological triplicates from HCT116 cells transfected with a control siRNA (siCTRL) or *ZMAT3* siRNAs (SMARTpool of 4 siRNAs). *ZMAT3* mRNA was strongly down-regulated, whereas

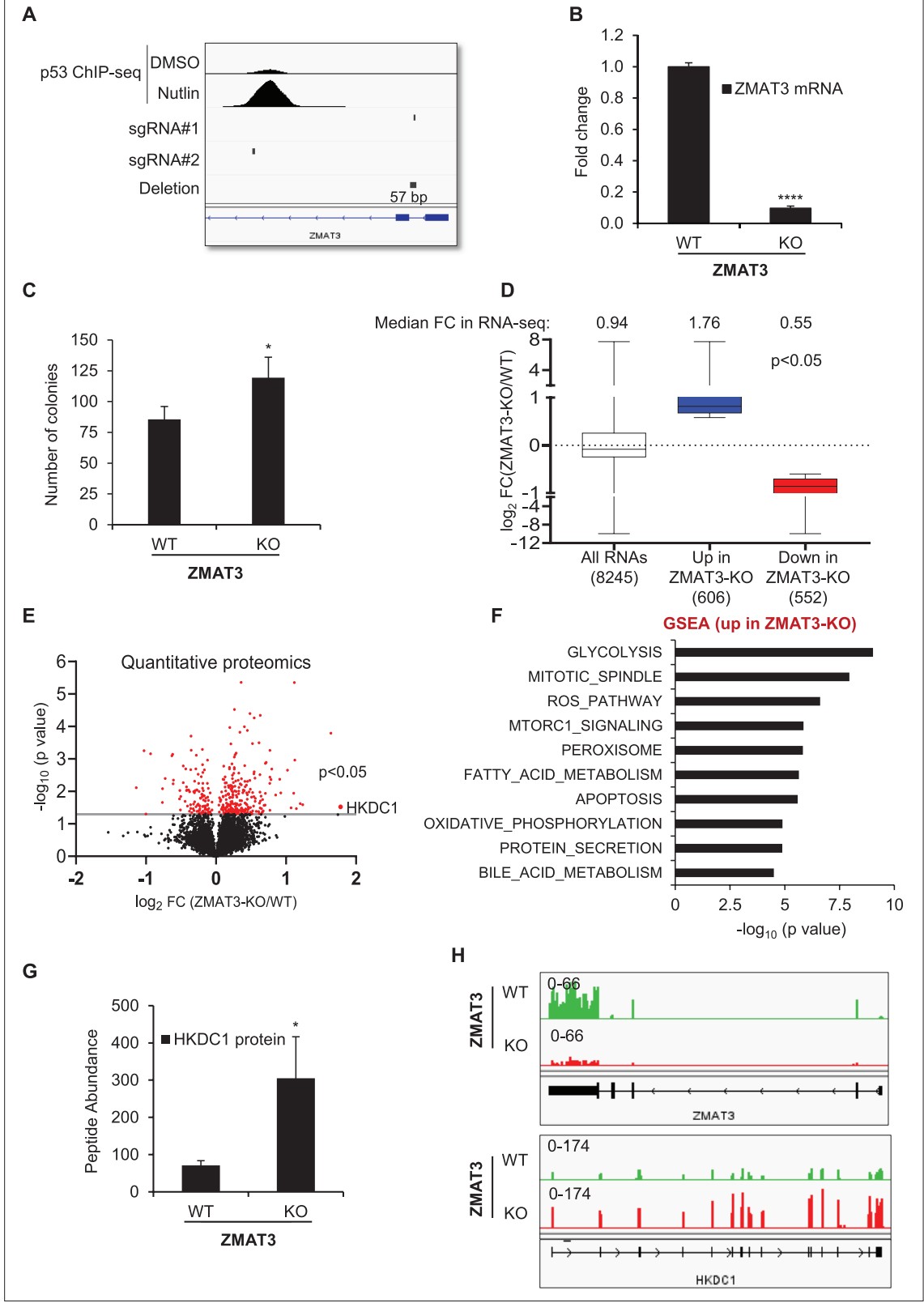

**Figure 1.** ZMAT3 depletion results in increased expression of genes related to glucose metabolism in colorectal cancer cells. (**A**) IGV snapshot showing the location of the two sgRNAs used to generate *ZMAT3*-KO HCT116 cells, the observed 57 bp deletion near sgRNA#2, and the p53 ChIP-seq peak in the *ZMAT3* locus in response to p53 activation upon Nutlin treatment. The p53 ChIP-seq data were previously published (***Andrysik et al., 2017***). (**B**) RT-qPCR analysis of *ZMAT3*-WT and *ZMAT3*-KO HCT116 cells from three biological replicates. *GAPDH* served as the housekeeping gene control. (**C**)

*Figure 1 continued on next page*

*Figure 1 continued*

Colony formation assays performed from *ZMAT3*-WT and *ZMAT3*-KO HCT116 cells in three biological replicates. (**D**) Notched box plot of the $\log_2$fold change (FC) in RNA abundance of differentially expressed genes from RNA-Seq of *ZMAT3*-KO and *ZMAT3*-WT HCT116 cells. Median values for each group are indicated at the top of each box, and the number of RNAs for which data were obtained for each group is indicated at the bottom. (**E**) Volcano plot showing differentially expressed proteins (shown in red) identified by global quantitative proteomics from *ZMAT3*-WT and *ZMAT3*-KO HCT116 cells. (**F**) Most significantly enriched pathways identified by GSEA of genes significantly upregulated ($p<0.05$) in the *ZMAT3*-KO versus *ZMAT3*-WT based on quantitative proteomics data. (**G**) TMT mass spectrometry peptide abundance of HKDC1 in *ZMAT3*-WT and *ZMAT3*-KO HCT116 cells. Values represent the average of five biological replicates for *ZMAT3*-WT and four biological replicates for *ZMAT3*-KO cells. (**H**) IGV snapshot showing *ZMAT3* and *HKDC1* transcripts from RNA-seq of *ZMAT3*-WT and *ZMAT3*-KO HCT116 cells. Error bars in panels B, C, and G represent SD from three independent experiments *$p<0.05$, ****$p<0.0001$.

The online version of this article includes the following source data and figure supplement(s) for figure 1:

**Figure supplement 1.** ZMAT3 loss increases proliferation and alters gene expression.

**Figure supplement 1—source data 1.** Uncropped immunoblots for *Figure 1—figure supplement 1A*.

**Figure supplement 1—source data 2.** Uncropped immunoblots for *Figure 1—figure supplement 1A*.

**Figure supplement 2.** ZMAT3 loss does not alter the levels of p53 and p21.

**Figure supplement 2—source data 1.** Uncropped immunoblots for *Figure 1—figure supplement 2*.

**Figure supplement 2—source data 2.** Uncropped immunoblots for *Figure 1—figure supplement 2A and B*.

*HKDC1* mRNA was modestly but significantly up-regulated (*Figure 2—figure supplement 1A* and *Supplementary file 3*), a result that was validated by RT-qPCR (*Figure 2C*). GSEA for the mRNAs up-regulated upon *ZMAT3* knockdown revealed that glycolysis was among the top 10 over-represented biological processes (*Figure 2—figure supplement 1B*). Additionally, comparison of the RNA-seq data from the *ZMAT3*-WT vs *ZMAT3*-KO HCT116 cells and CTRL siRNA vs *ZMAT3* siRNA trans-fected HCT116 cells indicated that 1023 genes were commonly up-regulated ($p<0.05$ and $\log_2$fold change >0), and 1042 genes were commonly down-regulated ($p<0.05$ and $\log_2$fold change <0) upon loss of ZMAT3 (*Figure 2—figure supplement 1C* and **D**), suggesting that *ZMAT3* depletion results in altered expression of thousands of genes. GSEA analysis for the top 500 mRNAs up-regulated upon *ZMAT3* knockdown showed that glycolysis was among the top 10 over-represented biological processes (*Figure 2—figure supplement 1E*). It should be noted that for the comparison of both RNA-seq datasets (*ZMAT3*-WT vs *ZMAT3*-KO and siCTRL vs. siZMAT3), we included genes that were consistently up- or down-regulated, without applying a fold change threshold, focusing instead on significantly altered genes ($p<0.05$) in both datasets. This allowed us to capture broader and more reproducible transcriptomic changes that occur upon *ZMAT3* depletion, including modest but signif-icant changes.

To determine if ZMAT3 inhibits *HKDC1* expression across diverse cell types, we performed RT-qPCR after *ZMAT3* knockdown in SW1222 (CRC cells), HCEC-1CT (immortalized human colonic epithelial cells), and HepG2 (liver cancer cells). We observed a significant increase in *HKDC1* mRNA levels upon *ZMAT3* knockdown in these cell lines (*Figure 2D*). At the protein level, we observed that HKDC1 levels increased upon *ZMAT3* knockdown in HCT116, HepG2, SW1222, and HCEC-1CT cells (*Figure 2E*, *Figure 2—figure supplement 1F-G*, *Figure 2—source data 1*, *Figure 2—figure supplement 1—source data 1*). To determine whether this regulatory relationship is conserved between human and mouse, we analyzed recently published RNA-seq data from *Zmat3* knockout mouse embryonic fibroblasts (MEFs) (*Boutelle et al., 2025*). *Hkdc1* mRNA was significantly up-regulated (~ 6-fold) in *Zmat3*-KO MEFs (*Figure 2F*). As expected, we observed a decrease in *Zmat3* mRNA, upregulation of the *Zmat3*-target gene *Mdm4* and no changes in *Trp53* mRNA levels in *Zmat3*-KO MEFs (*Figure 2F*).

In the context of human CRC, analysis of the TCGA colorectal adenocarcinoma (COAD) cohort revealed that *HKDC1* mRNA levels were significantly higher in tumors as compared to normal tissues (*Figure 2G*). Further analysis of the TCGA COAD data showed that, as compared to p53 wild-type tumors, mutant p53 tumors exhibited significantly higher *HKDC1* mRNA levels (*Figure 2—figure supplement 1H*). As expected for the p53-induced gene ZMAT3, and as shown previously in other cancer types (*Bieging-Rolett et al., 2020*), *ZMAT3* mRNA levels were significantly lower in the mutant p53 tumors compared to p53 wild-type tumors (*Figure 2—figure supplement 1I*). Furthermore, RNA-seq from *Trp53* knockout MEFs from the same study (*Boutelle et al., 2025*) showed significant upreg-ulation (~8.6 fold) of *Hkdc1* mRNA, and down-regulation of *Trp53* and its target genes *Zmat3* and

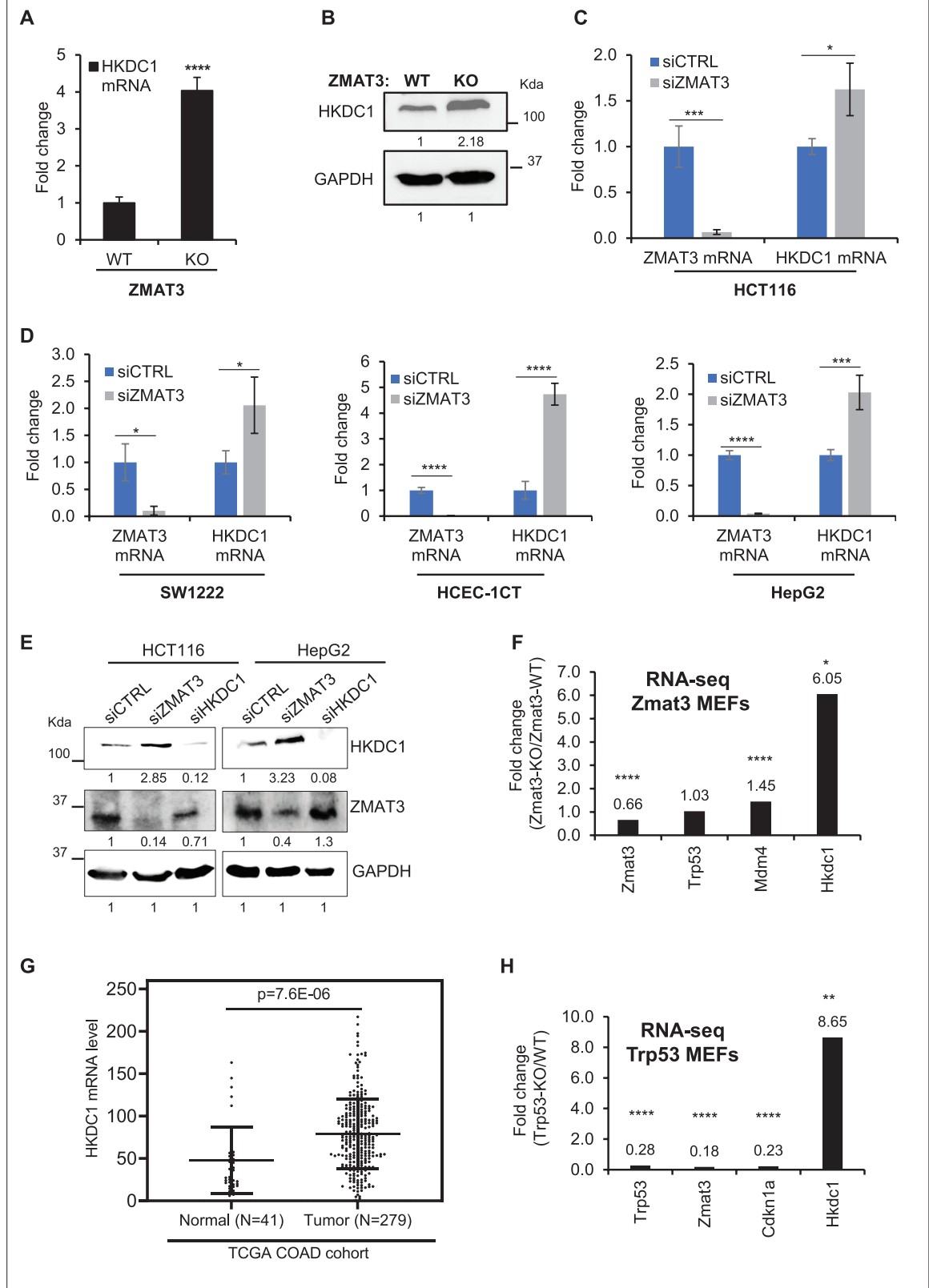

**Figure 2.** ZMAT3 negatively regulates *HKDC1* expression in diverse cell types. (**A, B**) RT-qPCR and immunoblotting for HKDC1 in *ZMAT3*-WT and *ZMAT3*-KO HCT116 cells. GAPDH served as the housekeeping gene control. RT-qPCR was performed in biological triplicates. (**C, D**) RT-qPCR analysis from the indicated cell lines in biological triplicates following transfection with control (CTRL) siRNA or *ZMAT3* siRNAs for 72 hr. *GAPDH* served as the housekeeping gene control. (**E**) Immunoblotting of whole-cell lysates from HCT116 and HepG2 cells after siRNA-mediated knockdown of *ZMAT3* or

*Figure 2 continued on next page*

*Figure 2 continued*

HKDC1 for 72 hr. GAPDH served as the loading control. (**F**) Fold change in *Zmat3, Trp53, Mdm4,* and *Hkdc1* mRNA levels from RNA-seq analysis of *Zmat3* knockout and wild-type mouse embryonic fibroblasts (MEFs). (**G**) Analysis of *HKDC1* mRNA levels in normal colon tissue and CRC samples from the TCGA COAD cohort. N indicates the number of samples in each group. (**H**) Fold change in *Trp53, Zmat3, Cdkn1a,* and *Hkdc1* mRNA levels from RNA-seq analysis of *Trp53* knockout and wild-type MEFs. Error bars in panels A, C, D, F, and H represent SD from three independent experiments. *$p<0.05$, **$p<0.01$, ***$p<0.001$, ****$p<0.0001$.

The online version of this article includes the following source data and figure supplement(s) for figure 2:

**Source data 1.** Uncropped immunoblots for *Figure 2B*.

**Source data 2.** Uncropped immunoblots for *Figure 2B and E*.

**Figure supplement 1.** *HKDC1* is a ZMAT3 repressed gene whose expression correlates with *TP53* mutation status in human CRC.

**Figure supplement 1—source data 1.** Uncropped immunoblots for *Figure 2—figure supplement 1F*.

**Figure supplement 1—source data 2.** Uncropped immunoblots for *Figure 2—figure supplement 1F and G*.

*Cdkn1a (p21)* (*Figure 2H*). Collectively, these data suggest that *ZMAT3* and *HKDC1* mRNA expression levels are negatively correlated within the p53 pathway, and inhibition of *HKDC1* expression by ZMAT3 and p53 is conserved between humans and mice.

## ZMAT3 inhibits mitochondrial respiration by downregulating HKDC1

Previous studies suggest that HKDC1 plays a crucial role in regulating glucose metabolism and proliferation in various cell types (*Khan et al., 2022*; *Zapater et al., 2022*). To determine whether ZMAT3 regulates glucose metabolism and/or proliferation by regulating *HKDC1* expression, we performed glucose metabolic assays. Since HKDC1 is a hexokinase that catalyzes the phosphorylation of glucose to glucose 6-phosphate upon entry into cells, we measured changes in hexokinase activity upon ZMAT3 depletion. We incubated the cells with the non-catabolic glucose analog 2-deoxy glucose (2-DG) for a short period of time and quantified the conversion of 2-DG to 2-DG6P using a luminescence-based assay. We observed a significant increase in relative 2-DG6P levels in *ZMAT3*-KO cells compared to WT cells, and interestingly, this increase was reversed upon *HKDC1* knockdown (*Figure 3A*). Furthermore, siRNA-mediated knockdown of *ZMAT3* in SW1222 and HepG2 also showed a similar increase in glucose uptake that was reversed upon simultaneous knockdown of *ZMAT3* and *HKDC1*, suggesting that ZMAT3 inhibits glucose uptake, and this effect is HKDC1-dependent (*Figure 3A*). We next determined if increased glucose uptake upon *ZMAT3* knockdown leads to increased glycolysis in cells. To do this, we performed Seahorse assays to measure extracellular acidification rate and calculated the glycolysis proton efflux. It should be noted that the proton efflux rate measured with Seahorse is reflecting the lactate production from glycolysis more than the pyruvate end point of glycolysis that fueled the mitochondria. Thus, we also measured mitochondria activity. Knockdown of *ZMAT3* or *HKDC1* by siRNAs resulted in modest, but not significant increases in basal glycolysis (*Figure 3B*).

Recent studies suggest that HKDC1 plays a significant role in regulating mitochondrial respiration, and depletion of *HKDC1* results in mitochondrial dysfunction and senescence (*Cui et al., 2024*; *Khan et al., 2022*). We, therefore, examined whether ZMAT3 regulates basal mitochondrial respiration by inhibiting *HKDC1* expression. To this end, we performed Seahorse assays to measure the oxygen consumption rate (OCR) following knockdown of *HKDC1* and/or *ZMAT3*. Interestingly, *ZMAT3* knockdown resulted in a significant increase in basal mitochondrial respiration, and simultaneous *ZMAT3* and *HKDC1* knockdown rescued this effect (*Figure 3C*), without altering oxygen consumption from non-mitochondrial respiration (*Figure 3—figure supplement 1*). These data indicate that *ZMAT3* regulates mitochondrial respiration without significantly affecting glycolysis. It is possible that mitochondria in *ZMAT3*-KO cells oxidize substrates other than those derived from glycolysis. Further studies will be required to determine these mechanisms in detail.

Because these phenotypes can be associated with proliferation, we next asked whether HKDC1 regulates proliferation and whether it is an effector of ZMAT3. Live cell proliferation assays and cell counting kit-8 (CCK-8) cell viability assays showed that knocking down *HKDC1* in *ZMAT3*-WT and *ZMAT3*-KO cells resulted in decreased proliferation, but the effect of *HKDC1* knockdown was more pronounced in *ZMAT3*-KO cells (*Figure 3D and E*). This observation was unexpected, given that HKDC1 acts downstream of ZMAT3. One possible explanation is that elevated *HKDC1* expression in *ZMAT3*-KO cells increases their reliance on HKDC1 for sustaining proliferation and that HKDC1

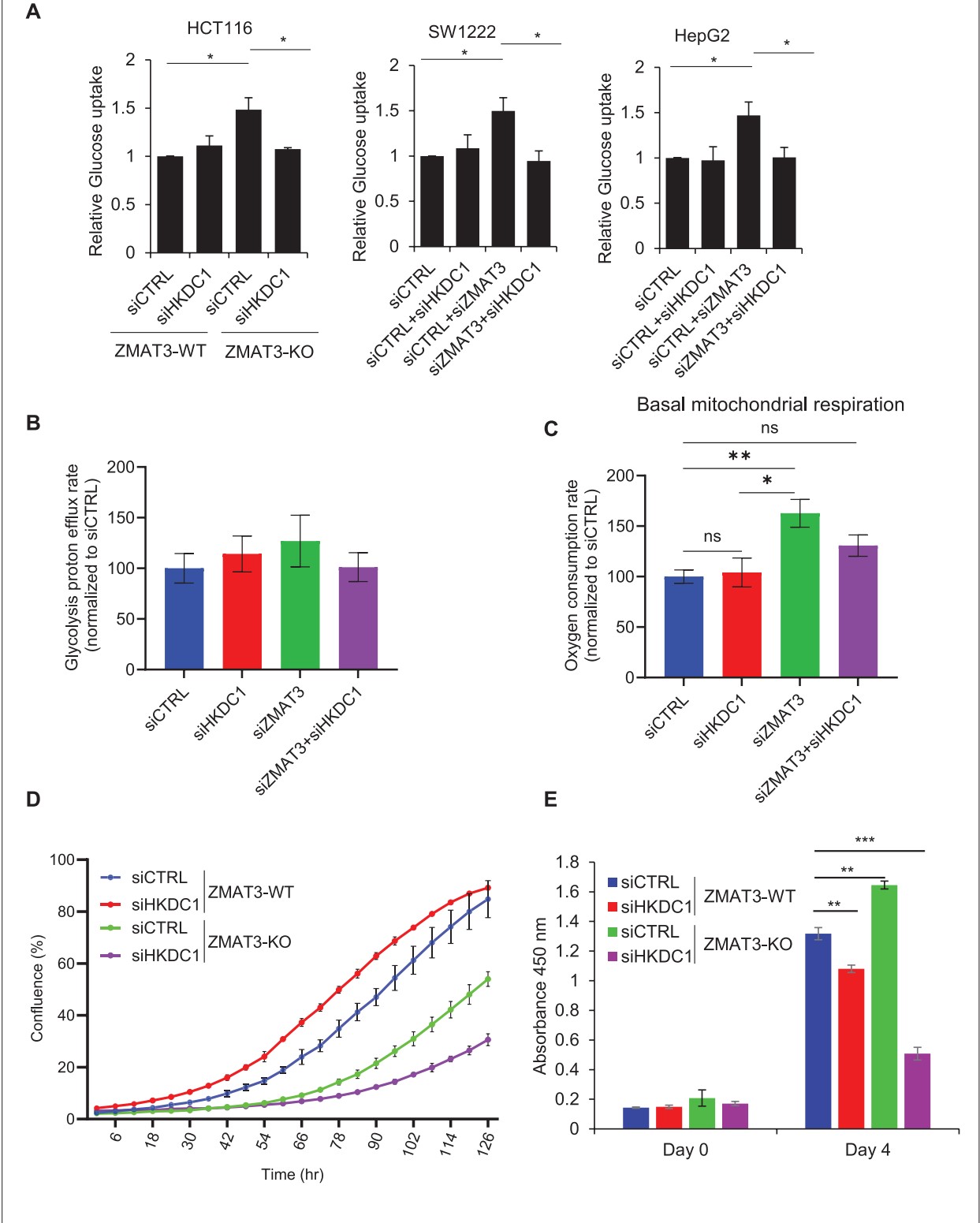

**Figure 3.** ZMAT3 inhibits mitochondrial respiration via HKDC1. (**A**) Glucose uptake was measured using a 2-deoxyglucose analog and a luminescence-based enzymatic assay in *ZMAT3*-WT and *ZMAT3*-KO HCT116 cells in the presence or absence of *HKDC1*. For SW122 and HepG2 cells, relative glucose uptake was measured following siRNA-mediated knockdown of *HKDC1* and/or *ZMAT3*. (**B, C**) Metabolic flux assays were performed to measure basal glycolysis rate and basal mitochondrial respiration rate in HCT116 cells after *ZMAT3* and/or *HKDC1* knockdown. (**D, E**) Incucyte live-cell proliferation assays and CCK8-based cell proliferation assays in *ZMAT3*-WT and *ZMAT3*-KO HCT116 cells in the presence or absence of siRNA-mediated *HKDC1*

*Figure 3 continued on next page*

*Figure 3 continued*

knockdown. Error bars in panels A, D, and E represent SD from three independent experiments, and error bars in panels B and C represent SD from four independent experiments. *$p<0.05$, **$p<0.01$, ***$p<0.001$.

The online version of this article includes the following figure supplement(s) for figure 3:

**Figure supplement 1.** Non-mitochondrial oxygen consumption in HCT116 cells following *ZMAT3* and/or *HKDC1* knockdown.

may also participate in additional pathways in *ZMAT3*-KO cells. Consequently, transient knockdown of *HKDC1* in *ZMAT3*-KO cells would have a more pronounced effect on proliferation due to their increased dependency on HKDC1 activity. In contrast, *ZMAT3*-WT cells, which express lower levels of *HKDC1* are less dependent on its function and, therefore, less sensitive to its depletion.

## The p53/ZMAT3 axis inhibits HKDC1 expression

Because *ZMAT3* transcription is activated by p53, we next examined whether the p53 pathway inhibits *HKDC1* expression. To this end, we performed RNA-seq on HCT116 cells transfected with siCTRL or s*ip53* (SMARTpool of four siRNAs). Interestingly, *HKDC1* mRNA was significantly up-regulated (~2.2 fold) upon *p53* knockdown (*Figure 4A and B* and *Supplementary file 4*). As expected, *p53* knockdown led to strong downregulation of mRNAs encoding *p53* and its target genes *p21* and *ZMAT3* (*Figure 4A and B*). We validated the observed up-regulation of *HKDC1* upon *p53* knockdown by RT-qPCR and immunoblotting (*Figure 4C and D*). In these experiments, mRNA and/or protein levels of p53, p21, and ZMAT3 were markedly decreased upon *p53* knockdown (*Figure 4C, D*, *Figure 4—source data 1*). At the transcriptome-wide level, ~2850 genes were differentially expressed ($p<0.05$) upon *p53* knockdown (*Supplementary file 4* and *Figure 4—figure supplement 1A*). Intersection of these differentially expressed genes ($p<0.05$) with those regulated by *ZMAT3* identified 351 genes commonly upregulated and 425 genes commonly downregulated genes (*Figure 4—figure supplement 1B and C*). GSEA revealed enrichment of glycolysis and the p53 pathway for the genes upregulated or downregulated following knockdown of *p53* or in *ZMAT3*-KO, respectively (*Figure 4—figure supplement 1D and E*).

Since *p53* knockdown resulted in increased *HKDC1* expression, we next asked whether increasing p53 expression would lead to downregulation of *HKDC1*. Indeed, analysis of our RNA-seq data revealed ~40% reduction in *HKDC1* mRNA levels following Nutlin treatment of siCTRL-transfected HCT116 cells (*Figure 4E* and *Supplementary file 3*). As expected for a direct p53 target gene, p53 activation by Nutlin led to upregulation of *ZMAT3* and *p21* mRNAs by ~2.5 and ~ 6-fold, respectively, while *p53* mRNA remained unchanged as Nutlin specifically stabilizes p53 protein levels (*Figure 4E*). At the protein level, p53 activation by Nutlin decreased HKDC1 levels and increased ZMAT3 and p21 levels in *ZMAT3*-WT cells (*Figure 4F*, *Figure 4—source data 1*). In *ZMAT3*-KO cells, basal HKDC1 levels were modestly higher, and Nutlin treatment resulted in a slight decrease (*Figure 4F*, *Figure 4—source data 1*). To further examine this regulation, we generated doxycycline-inducible ZMAT3-FLAG-HA expressing HCT116 cells to overexpress ZMAT3. Following 48 hr of doxycycline treatment, ZMAT3-FLAG-HA induction was confirmed by RT-qPCR and immunoblotting (*Figure 4G and H*, *Figure 4—source data 1*). Furthermore, knockdown of *p53* in these cells resulted in a significant increase in both *HKDC1* mRNA and protein levels under untreated conditions. Importantly, upon inducing ZMAT3 levels using doxycycline, the upregulation of *HKDC1* mRNA and protein associated with *p53* knockdown was no longer observed (*Figure 4I and J*, *Figure 4—source data 1*). Collectively, these data demonstrate that the p53/ZMAT3 axis plays a critical role in regulating *HKDC1* expression, with ZMAT3 acting downstream of p53 to inhibit *HKDC1* expression.

## ZMAT3 inhibits *HKDC1* transcription by interacting with the transcriptional activator JUN

ZMAT3 is an RNA-binding protein that regulates alternative splicing. We, therefore, hypothesized that ZMAT3 directly binds to the *HKDC1* pre-mRNA and regulates its splicing. However, in our previously published ZMAT3 PAR-CLIP (*Muys et al., 2021*), we did not detect binding of ZMAT3 to *HKDC1* pre-mRNA (data not shown). Furthermore, in RNA immunoprecipitation (RIP) assays performed in doxycycline-inducible ZMAT3-FLAG-HA HCT116 cells, we did not observe significant enrichment of *HKDC1* mRNA in the anti-FLAG RNA-IPs (*Figure 5—figure supplement 1A*). These data suggest that

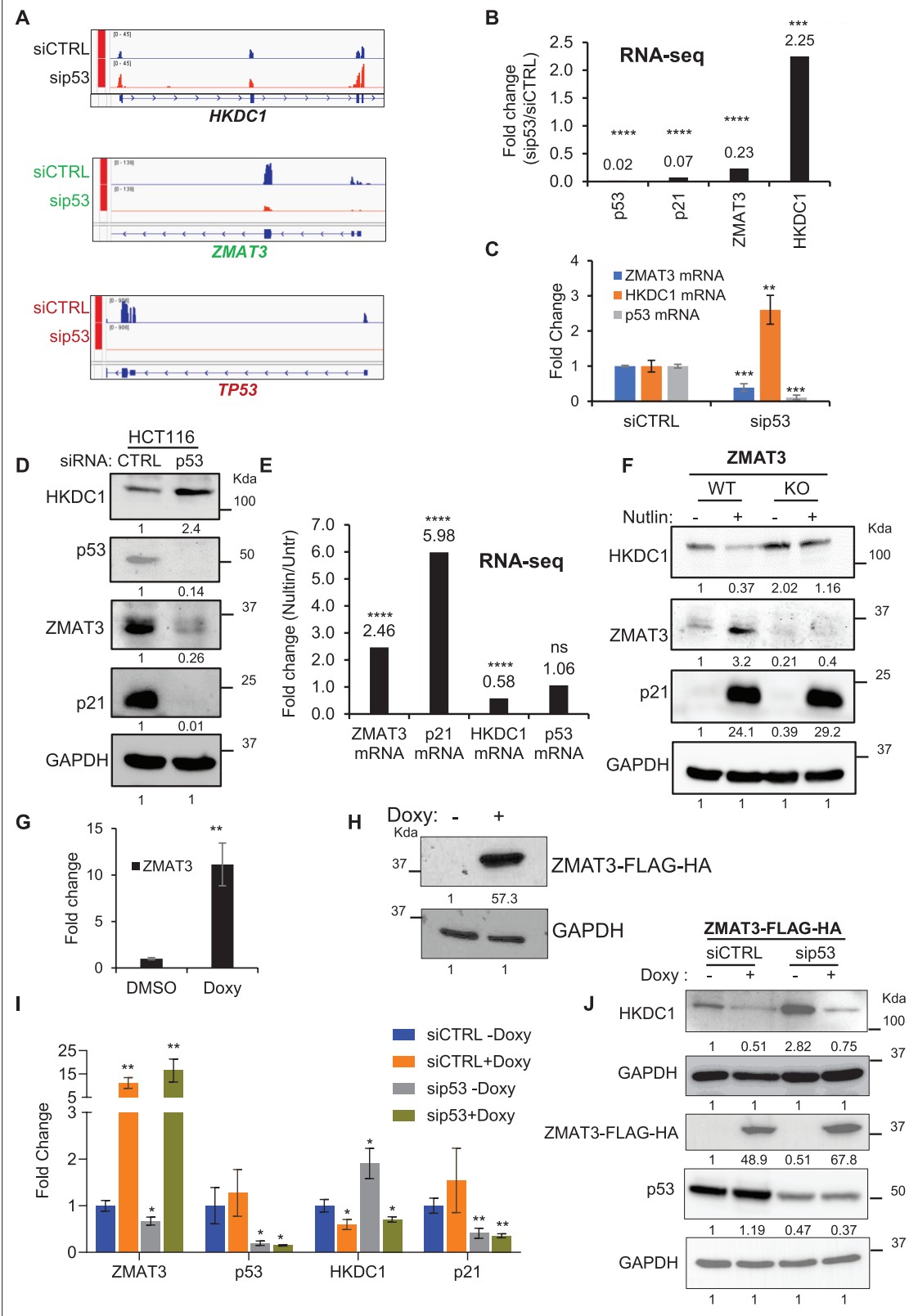

**Figure 4.** p53 negatively regulates *HKDC1* expression in a ZMAT3-dependent manner. (**A**) IGV snapshots from RNA-seq data following knockdown of *p53* using *p53* siRNAs in HCT116 cells. (**B**) Fold change for *p53, p21, ZMAT3,* and *HKDC1* mRNA levels from RNA-seq of HCT116 cells transfected with siCTRL and sip53. (**C, D**) HCT116 cells were transfected with siCTRL or *p53* siRNAs for 48 hr. *ZMAT3, p53,* and *HKDC1* mRNA or protein were measured by RT-qPCR (**C**) or immunoblotting of whole-cell lysates (**D**). *GAPDH* served as the housekeeping gene control. (**E**) Fold change in *ZMAT3, p21, HKDC1,*

*Figure 4 continued on next page*

*Figure 4 continued*

and *p53* mRNA levels from RNA-seq of HCT116 cells treated with DMSO or Nutlin for 6 hr. 'ns' denotes not significant. (**F**) Immunoblotting of whole-cell lysates from *ZMAT3*-WT and *ZMAT3*-KO HCT116 with or without Nutlin treatment for 24 hr. GAPDH served as the loading control. (**G, H**) Doxycycline (Doxy)-inducible ZMAT3-FLAG-HA HCT116 cells were treated with 2 μg/mL doxycycline for 48 h. *ZMAT3* mRNA and ZMAT3-FLAG-HA protein induction were measured by RT-qPCR (**G**) and immunoblotting using an anti-HA antibody (**H**). *GAPDH* served as the housekeeping control. (**I, J**) Doxycycline-inducible ZMAT3-FLAG-HA HCT116 cells were transfected with CTRL siRNA or *p53* siRNAs for 48 hr, followed by 48 hr of doxycycline treatment. *ZMAT3*, *p53*, and *HKDC1* mRNA and protein levels were measured by RT-qPCR (**I**) or immunoblotting from whole-cell lysates (**J**). *GAPDH* served as the housekeeping gene control. Error bars in panels B, C, E, G, and I represent SD from three independent experiments. *$p<0.05$, **$p<0.01$, ***$p<0.001$, ****$p<0.0001$.

The online version of this article includes the following source data and figure supplement(s) for figure 4:

**Source data 1.** Uncropped immunoblots for *Figure 4D, F, H and J*.

**Source data 2.** Uncropped immunoblots for *Figure 4D, F, H and J*.

**Figure supplement 1.** Transcriptomic overlap between p53 and ZMAT3 identifies shared pathways in HCT116.

ZMAT3 does not bind to *HKDC1* mRNA or pre-mRNA. Moreover, we analyzed changes in *HKDC1* pre-mRNA splicing using rMATS in HCT116 cells by reanalyzing our previously published RNA-seq data from siCTRL- and siZMAT3-transfected cells (*Muys et al., 2021*). We focused on splicing events with an adj.p $<0.05$ and a ΔPSI $> |0.1|$ (representing at least a 10% change in splicing). The splicing analysis did not reveal any significant alterations in *HKDC1* pre-mRNA splicing upon *ZMAT3* knockdown, suggesting that the observed increase in *HKDC1* mRNA is not at the level of splicing (data not shown).

Because ZMAT3 has three C2H2-zinc-finger motifs, it has the potential to bind DNA (*Figure 5—figure supplement 1B*; *Brayer and Segal, 2008*; *Gosztyla et al., 2024*; *Nabeel-Shah et al., 2024*). A recent study also reported that ZMAT3 and other zinc finger proteins function as dual DNA-RNA binding proteins (DRBPs) (*Gosztyla et al., 2024*). To determine whether ZMAT3 binds DNA and regulates *HKDC1* transcription directly, we performed ZMAT3-FLAG-HA CUT&RUN-seq and ChIP-seq in three biological replicates using anti-HA, anti-FLAG, or anti-ZMAT3 antibodies. However, we did not detect reproducible ZMAT3 binding at the *HKDC1* locus or at other genomic regions in HCT116 cells (data not shown).

We, therefore, hypothesized that ZMAT3 inhibits HKDC1 expression by interacting with a specific transcription factor. To identify proteins that interact with ZMAT3, we performed IPs using an anti-FLAG antibody followed by mass spectrometry on whole-cell lysates from untreated or doxycycline-treated ZMAT3-FLAG-HA HCT116 cells (*Figure 5A*, *Figure 5—figure supplement 1C* and *Figure 5—figure supplement 1—source data 1*). After filtering out common contaminants (<10% in CRAPome), this unbiased approach identified 21 ZMAT3-interacting proteins (*Figure 5B* and *Supplementary file 5*). As expected, ZMAT3 was the most strongly enriched protein (~36,000 fold). Of note, due to the low abundance of ZMAT3 in one of the control samples, the p-value for ZMAT3 enrichment was not highly significant ($p=0.09$) (*Supplementary file 5*).

Interestingly, the transcription factor JUN (c-Jun) was strongly enriched in the ZMAT3-FLAG pull-downs (~8500 fold). JUN is a proto-oncogene previously implicated in glucose metabolism (*Belgardt et al., 2010*; *Xiao et al., 2019b*) and p53 function (*Scherer et al., 2000*; *Eferl et al., 2003*). To further explore this potential regulatory link, we analyzed publicly available JUN ChIP-seq data from multiple ENCODE cell lines, focusing on JUN binding at the HKDC1 locus. Notably, in three cell lines, we found a JUN ChIP-seq peak containing the consensus JUN-binding motif within *HKDC1* intron 1. This peak coincided with ChIP-seq peaks for POLR2A, H3K27Ac, and H3K4Me3 in HCT116 cells (*Figure 5C*). We next validated the interaction between ZMAT3 and JUN by performing anti-FLAG IPs followed by immunoblotting on whole-cell lysates from untreated or doxycycline-treated ZMAT3-FLAG-HA HCT116 cells (*Figure 5D*, *Figure 5—source data 1*). Treatment with DNase or RNase abolished the interaction between ZMAT3 and JUN, indicating that nucleic acids mediate their association (*Figure 5—figure supplement 1D*, *Figure 5—figure supplement 1—source data 1*).

Importantly, *JUN* knockdown resulted in decreased *HKDC1* mRNA levels in *ZMAT3*-WT cells and rescued the elevated *HKDC1* mRNA and protein levels observed in *ZMAT3*-KO cells (*Figure 5E and F*, *Figure 5—source data 1*). To determine whether ZMAT3 inhibits JUN binding to the *HKDC1* locus, we performed ChIP-qPCR for JUN in *ZMAT3*-WT and *ZMAT3*-KO cells. JUN showed significant enrichment at the *HKDC1* intron 1 region compared to the IgG control in *ZMAT3*-WT cells, and this enrichment was further increased in *ZMAT3*-KO cells, indicating that ZMAT3 inhibits JUN binding to the

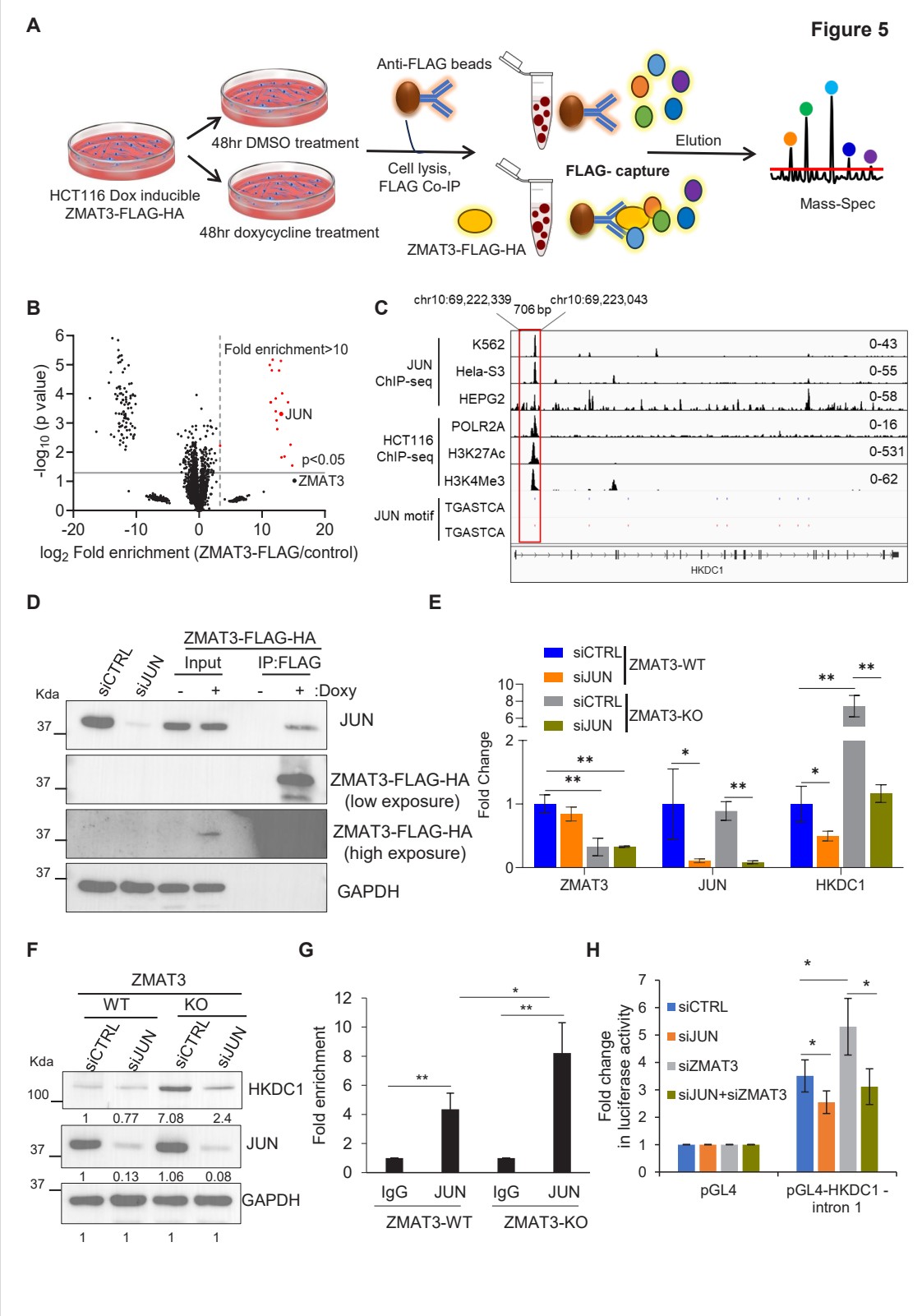

**Figure 5.** ZMAT3 inhibits *HKDC1* transcription by interacting with the transcription factor JUN. (**A**) Schematic of the workflow used to identify ZMAT3-FLAG-HA interacting proteins by IP-mass spectrometry in HCT116 cells expressing doxycycline-induced ZMAT3-FLAG-HA. (**B**) Volcano plot showing significantly enriched proteins (shown in red) identified by anti-FLAG IPs followed by mass spectrometry in the presence and absence of doxycycline in ZMAT3-FLAG-HA HCT116 cells. The vertical dotted line denotes a >10 fold enrichment cutoff. JUN was strongly enriched in the ZMAT3-FLAG IPs.

*Figure 5 continued on next page*

*Figure 5 continued*

(**C**) IGV snapshot showing JUN, POLR2A, H3K27Ac, and H3K4Me3 peaks at the *HKDC1* locus from ChIP-seq data from the ENCODE cell line datasets (accessions from top to bottom: ENCSR000FAH, ENCSR000EDG, ENCSR000EEK, ENCSR000EUU, ENCSR661KMA, and ENCSR333OPW). The JUN binding motif (TGASTCA) is shown in blue (positive strand) and in red (negative strand). (**D**) IP followed by immunoblotting using anti-FLAG beads and whole-cell lysates from untreated (no doxy) or doxy-treated ZMAT3-FLAG-HA HCT116 cells. Ten percent of cell lysate was used as input. GAPDH served as the loading control. (**E, F**) *ZMAT3*-WT and *ZMAT3*-KO HCT116 cells were transfected with CTRL siRNA or JUN siRNAs for 48 hr, followed by RT-qPCR (**E**) or immunoblotting of whole-cell lysates (**F**). GAPDH served as the housekeeping control. (**G**) JUN ChIP-qPCR was performed in biological triplicates from *ZMAT3*-WT and *ZMAT3*-KO HCT116 cells to determine the enrichment of JUN at *HKDC1* intron 1. (**H**) Luciferase assays were performed in biological triplicates following *JUN* and/or *ZMAT3* knockdown, and pGL4 or pGL4 construct containing the *HKDC1* intron 1 region. Error bars in panels E, G, and H represent SD from three independent experiments. $^*p<0.05$, $^{**}p<0.01$.

The online version of this article includes the following source data and figure supplement(s) for figure 5:

**Source data 1.** Uncropped immunoblots for *Figure 5D and F*.

**Source data 2.** Uncropped immunoblots for *Figure 5D and F*.

**Figure supplement 1.** ZMAT3 does not bind HKDC1 RNA, and its interaction with JUN is nucleic-acid dependent.

**Figure supplement 1—source data 1.** Uncropped immunoblots for *Figure 5—figure supplement 1C and D*.

**Figure supplement 1—source data 2.** Uncropped immunoblots for *Figure 5—figure supplement 1C and D*.

**Figure supplement 2.** Not all canonical JUN targets are regulated by ZMAT3.

**Figure supplement 3.** ZMAT3/JUN axis negatively regulates HKDC1 and some JUN target genes.

*HKDC1* intron 1 DNA (*Figure 5G*). We next cloned a ~700 bp DNA fragment encompassing the JUN and POLR2A binding peaks within *HKDC1* intron 1 into the pGL4 basic luciferase reporter vector, hereafter referred to as pGL4-HKDC1-intron 1. Knockdown of *JUN* in HCT116 cells significantly decreased luciferase activity of the *HKDC1* reporter (*Figure 5H*). Conversely, *ZMAT3* knockdown led to a marked increase in luciferase activity, which was rescued by simultaneous knockdown of *ZMAT3* and *JUN* (*Figure 5H*).

Next, to identify additional JUN targets that might be regulated by ZMAT3, we first examined a set of well-characterized JUN target genes (*GLS Lukey et al., 2016*, *SREBF1 Jin et al., 2022*, *SLC2A1 Zhu et al., 2022*, *CD36 Banerjee et al., 2025* ,and WEE1 *Kappelmann-Fenzl et al., 2019*). Using the ChIPAtlas dataset for human JUN and cross-referencing it with JUN peaks from three ENCODE cell lines, we found that only *GLS*, *SREBF1,* and *SLC2A1* showed consistent JUN binding in all three cell lines (*Figure 5—figure supplement 2A–E*). We then measured the expression of *GLS*, *SREBF1,* and *SLC2A1* by RT-qPCR in *ZMAT3*-WT and *ZMAT3*-KO cells, with or without JUN knockdown. *GLS* mRNA was significantly reduced upon JUN knockdown in both *ZMAT3*-WT and *ZMAT3*-KO cells but was not upregulated in *ZMAT3*-KO cells, indicating that *GLS* is regulated by JUN in HCT116 cells but is not regulated by ZMAT3. In contrast, SREBF1 and SLC2A1 expression remained unchanged upon JUN knockdown (*Figure 5—figure supplement 2F–H*). These data suggest that JUN targets are highly cell type specific and not all JUN targets are negatively regulated by ZMAT3. To move beyond this candidate approach and identify additional JUN targets potentially regulated by ZMAT3, we intersected the genes upregulated upon *ZMAT3* knockout (from our RNA-seq data) with the ChIP-Atlas dataset for human JUN and cross-referenced these with JUN peaks from three ENCODE cell lines. From this analysis, we selected for further analysis the top four candidate genes - *LAMA2, VSNL1, SAMD3,* and *IL6R* (*Figure 5—figure supplement 3A–D*). Like *HKDC1*, these genes were upregulated in *ZMAT3*-KO cells, and this upregulation was abolished upon siRNA-mediated *JUN* knockdown in *ZMAT3*-KO cells (*Figure 5—figure supplement 3E*). Moreover, by ChIP-qPCR, we observed increased JUN binding to the JUN peak for these genes in *ZMAT3*-KO cells as compared to the *ZMAT3*-WT (*Figure 5—figure supplement 3F*). These data suggest that the ZMAT3/JUN axis negatively regulates *HKDC1* expression and additional JUN target genes.

Collectively, these data suggest that *ZMAT3* plays a crucial role in regulating mitochondrial respiration and cell proliferation by suppressing *HKDC1* transcription. We propose a model in which, in *ZMAT3*-WT cells, p53 drives *ZMAT3* transcription, and ZMAT3 protein binds to JUN, thereby inhibiting JUN binding to the *HKDC1* locus and repressing *HKDC1* transcription. This repression maintains controlled mitochondrial respiration and proliferation. In the absence of *ZMAT3*, increased JUN binding at the *HKDC1* locus leads to elevated *HKDC1* expression, enhanced mitochondrial respiration, and increased proliferation (*Figure 6*).

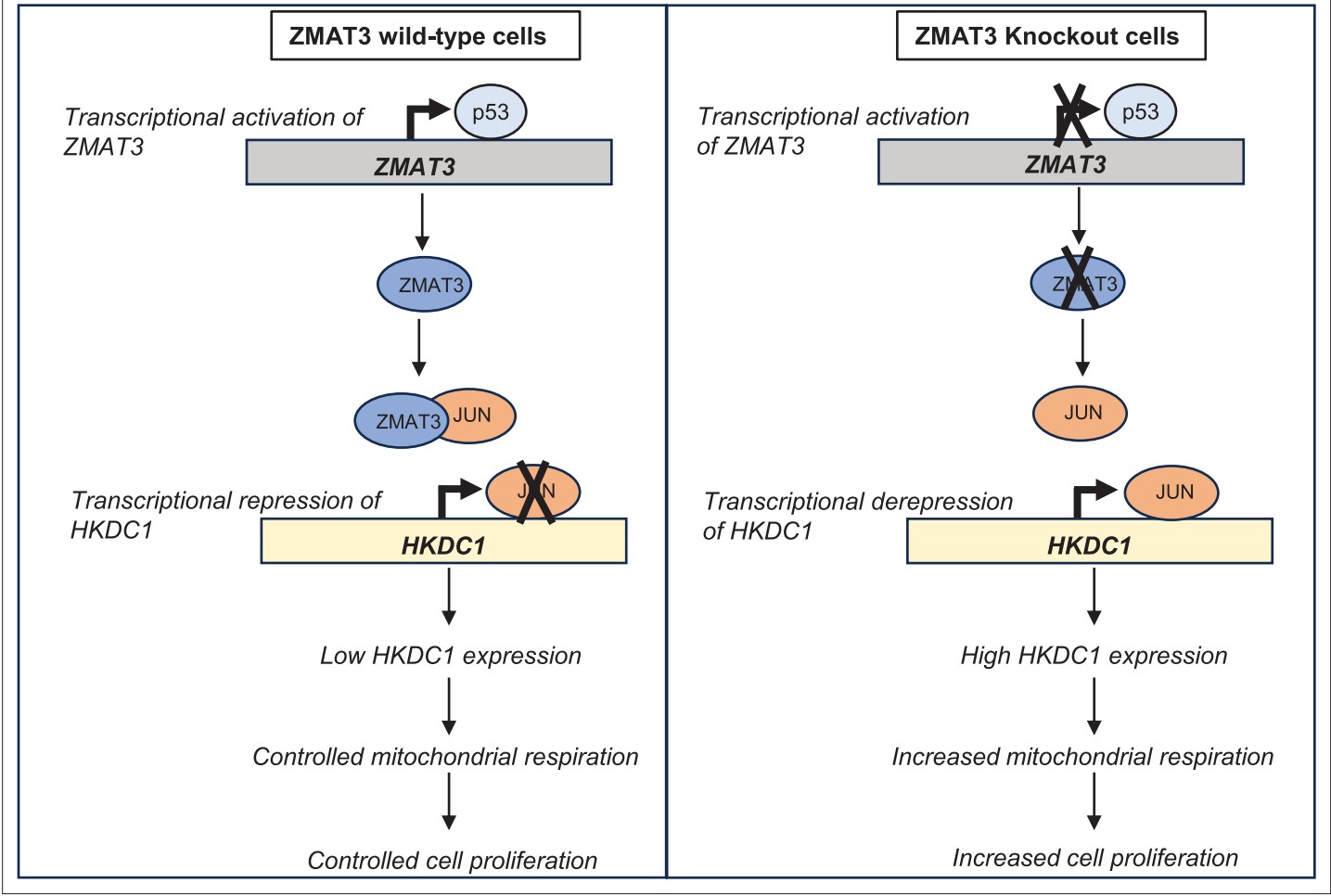

**Figure 6.** Model of ZMAT3-mediated regulation of *HKDC1* expression and mitochondrial respiration. In *ZMAT3*-WT cells, p53 activates *ZMAT3* transcription, leading to ZMAT3 protein binding to the transcription factor JUN. This interaction inhibits JUN binding to the *HKDC1* locus, resulting in low *HKDC1* expression and controlled mitochondrial respiration and cell proliferation. In *ZMAT3*-KO cells, JUN actively binds to the *HKDC1* locus and upregulates its expression, leading to increased mitochondrial respiration and enhanced cell proliferation.

## Discussion

Recent studies suggest that ZMAT3 significantly contributes to the tumor suppressive effects of p53 (*Bieging-Rolett et al., 2020*; *Bieging-Rolett and Attardi, 2021*). At the molecular level, ZMAT3 typically functions as an RNA-binding protein that acts as a key splicing factor and also regulates mRNA stability (*Bieging-Rolett et al., 2020*; *Muys et al., 2021*). Here, we unexpectedly found that transcription of *HKDC1*, the gene that is most strongly upregulated at the protein level in *ZMAT3*-deficient cells, is indirectly repressed by ZMAT3 through its interaction with the transcription factor JUN, thereby inhibiting JUN's binding to the *HKDC1* locus. Consistent with the established role of HKDC1 in glucose metabolism and mitochondrial respiration, ZMAT3 depletion led to increased mitochondrial respiration, a phenotype that was rescued by simultaneous knockdown of *ZMAT3* and *HKDC1*. These findings suggest that *HKDC1* is a key downstream effector of ZMAT3 in regulating cellular metabolism.

ZMAT3 has been known as a p53 target gene for more than two decades (*Hellborg et al., 2001*; *Varmeh-Ziaie et al., 1997*), yet its physiological functions and role in tumor suppression are only beginning to be understood. ZMAT3 belongs to the zinc finger family of proteins that play crucial roles in regulating gene expression through specific recognition of DNA sequences (*Cassandri et al., 2017*). Although primarily known for their involvement in transcription, these proteins have also been found to interact with RNA and proteins (*Kamaliyan and Clarke, 2024*; *Orth et al., 2021*). Among them, ZMAT3 is a member of the ZMAT domain-containing family that has three zinc fingers

of C2H2-type zinc fingers motif (*Font and Mackay, 2010*; *Krishna et al., 2003*). These domains play crucial roles in gene regulation by specifically binding to target molecules, such as DNA and RNA. A recent study (*Gosztyla et al., 2024*) revealed ZMAT3 as a DRBP (DNA- and RNA-binding protein), but in our hands using CUT&RUN-seq and ChIP-seq, we did not observe specific binding of ZMAT3-FLAG-HA or endogenous ZMAT3 to DNA (data not shown). In contrast, parallel, CUT&RUN-seq, and ChIP-seq assays for p53 performed exceedingly well (data not shown), confirming that the lack of ZMAT3 binding to DNA was not due to a technical limitation. It is possible that ZMAT3's interaction with DNA is cell type-specific or occurs under specific conditions, such as following DNA damage, but this requires further investigation.

Our findings that ZMAT3 interacts with and inhibits binding of JUN to the *HKDC1* locus provide mechanistic insights on how ZMAT3, without directly interacting with DNA, regulates *HKDC1* transcription. JUN is a protooncogene that plays a crucial role in both normal physiological processes and tumorigenesis by regulating cell proliferation, differentiation, senescence, and metastasis (*Eferl and Wagner, 2003*; *Zhou et al., 2017*; *Weiss and Bohmann, 2004*; *Spangler et al., 2011*). It functions as a transcription factor and a key component of the AP-1 complex and promotes RNA polymerase II-mediated transcription of target genes (*Lively et al., 2001*). Our data indicates that the ZMAT3–JUN interaction requires both DNA and RNA, indicatingthat ZMAT3 and JUN form RNA-dependent, chromatin-associated complexes. Although not investigated in our study, this aligns with emerging views that RBPs can function as chromatin-associated cofactors in transcription (*Xiao et al., 2019a*; *Kuninger et al., 2002*; *Wei et al., 2016*; *Ji et al., 2013*). For instance, functional interplay between transcription factor YY1 and the RNA binding protein RBM25 co-regulates a broad set of genes, where RBM25 appears to engage promoters first and then recruit YY1, with RNA proposed to guide target recognition (*Xiao et al., 2019a*). Future investigations are needed to determine the domains of JUN that interact with ZMAT3, in case there is direct interaction between these proteins, and what RNAs shape the ZMAT3 and JUN interaction within the genome. Although we did not detect chromatin binding of ZMAT3 with current reagents, improved ChIP-grade antibodies or endogenous epitope tagging (for ChIP-seq and CUT&RUN-seq) might clarify whether ZMAT3 occupies chromatin at physiological levels and how it modulates JUN binding to its target genes.

Because ZMAT3 regulates alternative splicing, which can lead to changes in protein levels without altering mRNA levels, in this study, we integrated global quantitative proteomics with RNA-seq data from *ZMAT3*-WT and isogenic *ZMAT3*-KO CRC cells. Furthermore, we integrated these data with RNA-seq from HCT116 cells upon *ZMAT3* or *p53* knockdown using siRNAs to make sure that the findings from the isogenic cell lines were not restricted to a single KO clone and to determine the role of the p53/ZMAT3 axis in regulating these genes. This approach identified HKDC1 as the most strongly upregulated protein upon ZMAT3 depletion. HKDC1 is emerging as an important regulator of tumor progression and is frequently upregulated in several cancers, including CRC (*Guo et al., 2015*; *Chen et al., 2019*; *Xu et al., 2021*). Besides its hexokinase activity, HKDC1 interacts with the mitochondrial membrane and plays an essential role in mitochondrial function (*Cui et al., 2024*; *Khan et al., 2022*). We further demonstrated that ZMAT3 suppresses glucose uptake and basal mitochondrial respiration by inhibiting *HKDC1* expression, leading to suppression of cell proliferation in CRC cells.

Our data also demonstrates that p53 negatively regulates *HKDC1* expression and this effect is ZMAT3-dependent, suggesting that the p53/ZMAT3/HKDC1 axis is an important component of the p53 network, specifically in the context of mitochondrial respiration and proliferation. The ability of p53 to induce cell cycle arrest and programmed cell death is important for tumor suppression (*Bieging et al., 2014*). However, p53 has several other functions that recent data strongly implicate in tumor suppression, particularly regarding the control of metabolism, such as glycolysis, mitochondrial respiration, and ferroptosis (*Vousden and Ryan, 2009*; *Tarangelo et al., 2018*). Metabolic reprogramming is a hallmark of cancer cells, which plays a pivotal role in cancer progression by providing energy and a wide variety of substrates for biosynthesis to support the rapid proliferation and survival of cancer cells (*Cairns et al., 2011*; *Pavlova and Thompson, 2016*; *Wolpaw and Dang, 2018*). p53 has been reported to play an important role in suppressing tumor development by regulating the expression and function of metabolic genes, directly *GLUT1* (*Schwartzenberg-Bar-Yoseph et al., 2004*), *GLUT4* (*Schwartzenberg-Bar-Yoseph et al., 2004*), *PFKFB3* (*Franklin et al., 2016*), and *PFKFB4* (*Ros et al., 2017*) or indirectly *HK2* (*Wang et al., 2014*), HIF1α (*Ravi et al., 2000*), and G6PD (*Jiang et al., 2011*). Our data uncovers HKDC1 as an indirect p53 target gene that is negatively regulated via ZMAT3.

Collectively, our findings provide key insights into the diverse functions of ZMAT3 and their involvement in gene regulation in the p53 pathway.

## Materials and methods

### Cell lines

HCEC-1CT, HepG2, HCT116, and HEK293T cells were purchased from the American Type Culture Collection (ATCC), and SW1222 cells were purchased from https://cancertools.org/. Cells were maintained in Dulbecco's Modified Eagle Medium (DMEM, Thermo Fisher Scientific, Catalog no. 11995065) supplemented with 10% fetal bovine serum (FBS; Thermo Fisher Scientific, Catalog no. 10082147) and 100 U/mL of penicillin and 0.1 mg/mL of streptomycin (Thermo Fisher Scientific, Catalog no. 15070063). Cultures were incubated at 37 °C in a humidified atmosphere containing 5% $CO_2$. All cell lines were regularly tested for mycoplasma using the Venor GeM Mycoplasma Detection Kit (Millipore Sigma, Catalog no. MP0025-1KT). All cell lines were authenticated by STR profiling.

### Targeted deletion of ZMAT3 using CRISPR/Cas9

To generate *ZMAT3*-KO clones, we employed the PiggyBac CRISPR/Cas9 system developed from the Zhang lab (*Shalem et al., 2014*). Two sgRNAs flanking the p53RE within *ZMAT3* intron 2 were designed and individually cloned in pENTR221 vector. These constructs were electroporated into $1 \times 10^6$ parental HCT116 cells using the Amaxa Cell Line Nucleofector Kit (Lonza, Catalog no. VCA-1005), together with pT3.5-FLAG-Cas9, pCDNA-pB7, and pBSB-CG-LUC-GFP-(puro)(cre+) vectors. After two days, cells were treated with 2 μg/mL puromycin (Thermo Fisher Scientific, Catalog no. A1113803) for three days. Following puromycin selection, single cells were seeded into 96-well plates to isolate *ZMAT3*-WT and *ZMAT3*-KO clones. Clones were expanded for three weeks and transferred to 24-well plates. Total RNA was isolated from each well, and *ZMAT3* expression was measured by RT-qPCR and normalized to *GAPDH*. Genomic DNA was extracted from individual clones showing a strong reduction of *ZMAT3* expression, and the genomic region flanking the p53RE of *ZMAT3* was PCR-amplified and verified by Sanger sequencing.

### Plasmid construction, lentivirus production, and siRNA transfections

The pLVX-Puro-Tet-One vector from TaKaRa (Catalog no. 631849) was used as a backbone to construct an expression construct encoding ZMAT3-3xFLAG-2xHA. The resulting plasmids were transformed into *E. coli* DH5α cells (Thermo Fisher Scientific, Catalog no. 18265017), and purified using the Monarch Plasmid Miniprep Kit (NEB, Catalog no. T1010L). Lentiviral particles were produced in $3 \times 10^5$ HEK293T cells after co-transfection of 1 μg of plasmid DNA and a third-generation lentiviral packaging system using Lipofectamine 2000 (Thermo Fisher Scientific Catalog no. 11668027). HCT116 cells were transduced at a multiplicity of infection (MOI) of ~1 and, after 2 days, were selected with 2 μg/mL puromycin (Thermo Fisher Scientific, Catalog no. A1113803) for 1 week.

For luciferase assays, a 706 bp genomic region encompassing the JUN-binding peak within the *HKDC1* locus was cloned into the pGL4-Basic vector (Promega, Catalog no. E6651), generating the pGL4-HKDC1-intron 1 reporter. To do this, a gene fragment corresponding to chr10:69,222,339–69,223,043 (hg38) was synthesized by Twist Bioscience and included restriction sites for KpnI (NEB, Catalog no. R3142) and XhoI (NEB, Catalog no. R0146S) at 5' and 3' ends, respectively. Both the pGL4-Basic vector and insert DNA fragment were digested with these enzymes and purified using the Monarch DNA Gel Extraction Kit (NEB, Catalog no. T1020S) or QIAquick PCR purification kit (Qiagen, Catalog no. 28106), and ligated using T4 DNA ligase (NEB, Catalog no. M0202S) to generate the final pGL4-HKDC1-intron 1 luciferase reporter.

### siRNA transfections

Reverse transfections were performed using Lipofectamine RNAiMAX Transfection Reagent (Thermo Fisher Scientific Catalog, no.13778075) and Opti-MEM (Thermo Fisher Scientific, Catalog no. 31985062) in HCT116, HCEC-1CT, HepG2, and SW1222 cells according to the manufacturer's protocol. The final concentration of siRNAs was 20 nM. For RT-qPCR and immunoblotting, two rounds of transfection were performed: the second transfection was performed 48 hr after the first transfection and cells were harvested after 72 hr. *HKDC1* siRNA transfections were conducted for only

one round following the first transfection with siCTRL or si*ZMAT3*. Negative Control siRNA (Qiagen, Catalog no. 1027281) was used as a control. The following SMARTpool siRNAs were used: si*ZMAT3* (Horizon Discovery, Catalog no. L-017382-00-0005), si*JUN* (Horizon Discovery, Catalog no. L-003268-00-0005), and si*p53* (Horizon Discovery, Catalog no. L-003329-00-0005). For metabolic assays, cells were transfected with siRNAs targeting more than one gene (e.g. si*HKDC1* and si*ZMAT3*) at 20 nM each. HCT116 cells expressing ZMAT3-FLAG-HA were transfected for 48 hr, reseeded, and treated with 2 µg/mL doxycycline to induce ZMAT3-FLAG-HA expression.

## Luciferase assays

HCT116 cells were transfected with siRNAs for 48 hr, and then $1 \times 10^5$ cells were reseeded in 24-well plates for luciferase assays. The following day, cells were co-transfected with 250 ng of pGL4 or pGL4-HKDC1-intron 1 and 25 ng of pRL-TK (Promega, Catalog no. E2231), together with 20 nM of either AllStars Negative Control siRNA, si*JUN* or si*ZMAT3*. Lipofectamine 2000 transfection reagent (Thermo Fisher Scientific, Catalog no. 11668027) was used for the co-transfections, according to the manufacturer's protocol. After 2 days, firefly and Renilla luciferase activities were measured using the Dual-Luciferase Reporter Assay System (Promega, Catalog no. E1910) on an EnSight Multimode plate reader (PerkinElmer). Firefly luminescence values were normalized to Renilla luminescence to account for transfection efficiency.

## RNA extraction and RT-qPCR

Cells were washed with 1 x DPBS (Thermo Fisher Scientific, Catalog no. 14190250) and total RNA was isolated using 500 µL of TRIzol Reagent (Thermo Fisher Scientific, Catalog no. 15596018). To prepare cDNA, 500 ng of total RNA was reverse-transcribed using the iScript Reverse Transcription Supermix (Bio-Rad, Catalog no. 1708841). Quantitative PCR (qPCR) was performed using 2.5 µL of diluted cDNA combined with 5 µL of 2 x FastStart Universal SYBR Green Master (Rox) (Millipore Sigma, Catalog no. 4913914001) and 0.5 µM (final concentration) of each primer in a 10 µL total reaction volume on a StepOnePlus Real-Time PCR machine (Applied Biosystems). Fold change was calculated using the $2^{-\Delta\Delta Ct}$ method, with *GAPDH* mRNA serving as housekeeping control.

## Immunoblotting

For immunoblotting, cells were lysed in 1 mL of RIPA buffer (Thermo Fisher Scientific, Catalog no. 89901). Lysates were sonicated three times for 5 s each at 50% amplitude using a VirTis VIRSONIC 100 sonicator and centrifuged at 16,000 × g for 10 min at 4 °C. The supernatant was collected, and protein concentration was determined using the Pierce BCA Protein Assay Kit (Thermo Fisher Scientific, Catalog no. 23225). For SDS-PAGE, 20–50 µg of total protein was loaded per lane and transferred to a PVDF membrane using a Bio-Rad semi-dry transfer apparatus. Membranes were blocked for 1 hr with TBST (Tris-Buffered Saline: 19.98 mM Tris, 136 mM NaCl, and 0.05% Tween, pH 7.4) containing 5% skim milk and then incubated with the primary antibody overnight at 4 °C. The following primary antibodies were used: anti-FLAG (1:1000 dilution; Sigma, Catalog no. F1804), anti-p53 (DO-1) (1:1000 dilution; Santa Cruz Biotechnology, Catalog no. sc-126), anti-ZMAT3 (1:500 dilution; Santa Cruz Biotechnology, Catalog no. sc-398712), anti-JUN (1:1000 dilution; Cell Signaling, Catalog no. 9165 S), and anti-HKDC1 (Proteintech, Catalog no: 25874–1-AP). Anti-GAPDH antibody (1:6000 dilution; Cell Signaling, 5174 S) was used for loading control. After incubation with HRP-conjugated secondary antibody (1:5000 dilution) for 1 hr at room temperature, the immunoblot was developed using the ECL Prime Western Blotting Detection Reagent (Fisher Scientific, Catalog no. RPN2232). Band intensities were quantified using ImageJ software and normalized to GAPDH.

## Colony formation assays

One thousand *ZMAT3*-WT and *ZMAT3*-KO HCT116 cells were seeded into 6-well plates. After 12 days, colonies were fixed with ice-cold methanol for 15 min and stained with 0.5% crystal violet (prepared in 10% methanol) for 15 min. Images were captured, and colony coverage area was quantified using ImageJ software.

## Incucyte proliferation assays

For proliferation assays, 1000 cells were seeded per well in a 96-well plate. Cells were incubated in an Incucyte S3 Live-Cell Analysis System (Sartorius) and imaged every 6 hr for 5 days. Images were analyzed using the manufacturer's software to determine percent confluence over time.

## Cell viability assays

To determine cell viability, cells were incubated with Cell Counting Kit-8 (CCK-8; Dojindo, Kumamoto, Japan) for 4 hr, and absorbance at 450 nm was measured using an Envision microplate reader (PerkinElmer).

## Glucose uptake assays

siCTRL, si*ZMAT3,* and si*HKDC1* were transfected into HCT116, SW1222, and HepG2 cells seeded in poly-L-lysine–coated white 96-well plates with opaque bottoms (Costar) and incubated at 37 °C for 24 hr. After the incubation, the growth medium was removed, and cells were washed twice with PBS to eliminate residual glucose. Glucose uptake was measured using the Glucose Uptake-Glo Assay Kit (Promega, Catalog no. J1341) according to the manufacturer's instructions. Uptake was initiated by adding 1 mM 2-deoxyglucose and incubating for 10 min at 37 °C, after which luminescence was recorded using an Envision Multimode Plate Reader (PerkinElmer).

## Metabolic flux assays

First, HCT116 cells were reverse-transfected with siCTRL or siZMAT3 for 48 hr. After 48 hr, a second round of transfection was conducted in 24-well Seahorse plates (Agilent Technologies Inc, Catalog no. 100777–004,) at a concentration of $3 \times 10^4$ per 200 µl per well. After 48 hr cells were washed twice with 500 mL of XF DMEM Medium, pH 7.4 (Agilent Technologies Inc, Catalog no. 103575–10) containing 1 mM pyruvate, 2 mM of glutamine, and 10 mM of glucose (Agilent Technologies Inc, Catalog no. 103578–100, 103579–100, 103577–100). Eventually, cells were incubated for 45 min in a non-CO2 incubator prior to the assays. Meanwhile, drugs from the Mito Stress Test Kit (Agilent Technologies Inc, Catalog no. 103015–10) and Glycolytic Rate Assay Kit (Agilent Technologies Inc, Catalog no. 103344–100) were prepared in XF DMEM Medium, pH 7.4 containing 1 mM pyruvate, 2 mM of glutamine and 10 mM of glucose. For the Mito Stress Test Kit, the working drug solutions concentration were Oligomycin 1.5 µM, FCCP 1.0 µM and Rot/AA 0.5 µM. For the glycolytic Rate Assay Kit, the working drug solutions concentration were Rot/AA 0.5 µM and 2-DG 50 mM. After the 45 min incubation, the plates were loaded into the Seahorse Analyzer and the commercial protocols for the drugs distribution were used. Immediately after the assays, media was removed from wells and cells frozen for later protein concentration measurement. Protein concentrations were measured with Pierce BCA Protein Assay kit (Thermo Fisher Scientific, Catalog no. 23225) and used to normalize Seahorse results.

## Gene set enrichment analysis

GSEA was performed using the MSigDB Hallmark gene sets from the Molecular Signature database (https://www.gsea-msigdb.org/gsea/msigdb/human/annotate.jsp).

## Co-immunoprecipitation

ZMAT3-FLAG-HA coimmunoprecipitations were performed using anti-FLAG M2-coated magnetic beads (Sigma-Aldrich, Catalog no. M8823). Approximately $5 \times 10^7$ doxycycline-treated or untreated HCT116 ZMAT3-FLAG-HA cells were lysed in IP lysis buffer (10 mM Tris/Cl pH 7.5, 150 mM NaCl, 0.5 mM EDTA, 0.5% NP40) supplemented with 1 mM PMSF and complete protease inhibitor cocktail (Roche). Lysates were incubated for 30 min at 4 °C with gentle mixing and clarified by centrifugation at 16,000 × g for 10 min at 4 °C. Equal amounts of clarified lysates were incubated with prewashed M2 beads overnight at 4 °C with constant rotation. Beads were magnetically separated from the unbound material and washed four times with IP wash buffer (10 mM Tris/Cl pH 7.5, 150 mM NaCl, 0.05% NP40, 0.5 mM EDTA). Bound proteins were eluted with FLAG elution buffer containing 125 µg/mL 3x FLAG peptide (Sigma-Aldrich, Catalog no. F4799). Equal volumes of eluates were boiled at 100 °C for 5 min in Laemmli sample buffer and then centrifuged at 16,000 × g for 5 min at room temperature. Total cell lysate was used as input, and proteins were detected by immunoblotting. For IP-mass spectrometry, bead-bound samples were processed directly after washing the beads, without the elution step.

## RNA-immunoprecipitation

ZMAT3-FLAG-HA RNA-immunoprecipitations were performed using anti-FLAG M2-coated magnetic beads (Sigma-Aldrich Catalog no. M8823), as described above for the co-IP assays. After the IPs, beads were magnetically separated from the unbound material and washed four times with IP Wash buffer. Bound RNAs were extracted directly from the beads using TRIzol Reagent (Thermo Fisher Scientific, Catalog no. 15596018) following the manufacturer's protocol.

## ChIP-qPCR

ChIP-qPCR was performed using the ChIP-IT Express Kit (Active Motif, Catalog no. 53008) following the manufacturer's instructions. Briefly, $5 \times 10^7$ *ZMAT3*-WT and *ZMAT3*-KO HCT116 cells grown in 15 cm plates were cross-linked with 1% formaldehyde, scraped, lysed, and sheared. Chromatin fragment size was verified on a 1% agarose gel. Chromatin was immunoprecipitated overnight at 4 °C using 1 μg of anti-JUN antibody or IgG isotype control. The IP material was washed, eluted, and reverse crosslinked overnight at 65 °C. ChIP DNA was purified using the QIAquick PCR Purification Kit (Qiagen, Catalog no. 28104) and analyzed by qPCR. ChIP-qPCR primers were designed based on the genomic regions harboring JUN-binding peaks in *HKDC1*, *LAMA2*, *VSNL1*, *SAMD3*, and *IL6R* (*Figure 5C*, *Figure 5—figure supplement 2A-D*).

## RNA-seq and analysis

RNA-seq was performed in biological triplicates from HCT116 *ZMAT3*-WT and *ZMAT3*-KO cells. Total RNA was isolated using the RNeasy Plus Mini Kit (Qiagen, Catalog no. 74134) following the manufacturer's instructions. Libraries were prepared using the Illumina Stranded mRNA Ligation Library Kit with 450 ng of total RNA as the input for mRNA capture using oligo(dT)-coated magnetic beads. The captured mRNA was fragmented and reverse-transcribed using random primers to synthesize first-strand cDNA, followed by second-strand synthesis. The resulting double-stranded cDNA was subjected to standard Illumina library preparation, including end repair, adapter ligation, and PCR amplification, to generate sequencing-ready libraries. The final purified libraries were quantified by qPCR prior to cluster generation and paired-end sequencing on the NovaSeq 6000 platform using an SP 200-cycle kit. Demultiplexing and conversion of binary base call (BCL) files to FASTQ format were performed using Illumina bcl2fastq v2.20. Sequencing reads were trimmed to remove adapters and low-quality bases using Cutadapt (v1.18). Trimmed reads were aligned to the human reference genome (hg38) using the STAR aligner (v2.7.0f) with the two-pass alignment option and GENCODE annotation (v30). Gene and transcript quantification was performed using RSEM (v1.3.1) based on GENCODE annotation.

## Mass spectrometry sample preparation

For total protein identification, cell pellets were suspended in 8 M urea buffer supplemented with protease and phosphatase inhibitors (Roche). All samples were transferred to 2 mL TissueLyser tubes containing 5 mm steel balls and kept on ice. Cells were lysed in a TissueLyser (Qiagen) for 2×2 min, with chilling at −20 °C for 2–3 min between cycles. Lysates were centrifuged at 12,500 × g for 15 min at 4 °C, and supernatants were transferred to new tubes. Protein concentrations were measured using the BCA assay (Thermo Fisher Scientific). For downstream processing, 200 μg of protein from each sample was reduced with 10 mM DTT at 56 °C for 1 hr and alkylated with 20 mM iodoacetamide at room temperature for 30 min in the dark**.** Following alkylation, samples were diluted fourfold with 50 mM triethylammonium bicarbonate (TEAB) to reduce the urea concentration to 2 M and digested with trypsin (substrate:enzyme = 40:1) at 37 °C overnight. Digested peptides were desalted using C18 columns and lyophilized. Peptide concentrations were measured using the colorimetric BCA peptide assay (Thermo Fisher Scientific). For TMT labeling, 100 μg of digested peptides was labeled with TMTpro 16-plex reagent at room temperature in the dark for 1 hr. Reactions were stopped by the addition of 5% hydroxylamine and incubation at room temperature in the dark for 15 min. Following labeling, peptide samples were pooled and lyophilized. The pooled samples were reconstituted in 0.1% TFA and fractionated using the high-pH reverse-phase peptide fractionation kit (Thermo Fisher Scientific) with nine elution buffers containing 0.1% triethylamine and 10%, 12.5%, 15%, 17.5%, 20%, 22.5%, 25%, 50%, and 75% acetonitrile, respectively. Fractions were lyophilized separately.

For ZMAT3-FLAG interacting protein identification, IP samples were solution digested with trypsin using S traps (Protifi), following the manufacturer's instructions. Briefly, proteins were denatured in 5% SDS, 50 mM triethylammonium bicarbonate (TEAB) pH 8.5. They were next reduced with 5 mM Tris(2-carboxyethyl)phosphine (TCEP) and alkylated with 20 mM iodoacetamide. The proteins were acidified to a final concentration of 2.5% phosphoric acid and diluted into 100 mM TEAB pH 7.55 in 90% methanol. They were loaded onto the S-traps, washed four times with 100 mM TEAB pH 7.55 in 90% methanol, and digested with trypsin overnight at 37 °C. Peptides were eluted from the S-trap using 50 mM TEAB pH 8.5; 0.2% formic acid in water; and 50% acetonitrile in water. The elutions were pooled and dried by lyophilization.

## Mass spectrometry analysis

Dried peptides were resuspended in 5% acetonitrile, 0.05% TFA in water for mass spectrometry analysis on an Orbitrap Exploris 480 (Thermo) mass spectrometer. The peptides were separated on a 75 µm × 15 cm, 3 µm Acclaim PepMap reverse phase column (Thermo) at 300 nl/min using an UltiMate 3000 RSLCnano HPLC (Thermo) and eluted directly into the mass spectrometer. For analysis, parent full-scan mass spectra acquired at 120,000 FWHM resolution and product ion spectra at 45,000 resolution with a 0.7 m/z isolation window. Proteome Discoverer 3.0 (Thermo) was used to search the data against the human database from Uniprot using SequestHT and with INFERYS rescoring. The search was limited to tryptic peptides, with maximally two missed cleavages allowed. Cysteine carbamidomethylation and TMT pro modification of lysine and peptide N-termini were set as a fixed modification, with methionine oxidation as a variable modification. The precursor mass tolerance was 10 ppm, and the fragment mass tolerance was 0.02 Da. The Percolator node was used to score and rank peptide matches using a 1% false discovery rate. TMT quantitation was performed using the Reporter Ions Quantifier nodes with correction of the values for lot-specific TMT reagent isotopic impurities.

## TCGA COAD gene expression

Gene expression and clinical data for COAD were obtained from The Cancer Genome Atlas (TCGA) using the TCGAbiolinks package (v2.16.0) in R (v4.0.0). The GDC query function was used to query samples by setting the data category to 'gene expression,' data type to 'gene expression quantification,' platform to 'Illumina HiSeq,' file type to 'normalized results,' and experimental strategy to 'RNA-seq.' Data were then downloaded using GDC download, followed by GDC prepare to obtain normalized gene expression values.

## Statistical analysis

Statistical analysis for all data was performed using data from at least three independent replicates. The difference between two groups were determined using a two-tailed Student's t-test, while comparisons involving multiple groups were analyzed using two-way ANOVA.

## Acknowledgements

We thank the CCR Genomics Core, CCR, NCI, Bethesda, MD, for valuable assistance with Sanger sequencing and Agilent TapeStation. We also thank the CCR Sequencing Facility, NCI, Frederick, MD, for performing the RNA-seq. Finally, we thank the members of the Lal lab for discussion and suggestions. This research was supported by the Center for Cancer Research, National Cancer Institute (NCI), National Institutes of Health (NIH) Intramural Research Program (project number ZIA BC011646) and federal funds from the NCI, NIH under contract 75N91019D00024. The contributions of the NIH author(s) were made as part of their official duties as NIH federal employees, are in compliance with agency policy requirements, and are considered Works of the United States Government. However, the findings and conclusions presented in this paper are those of the author(s) and do not necessarily reflect the views of the NIH or the U.S. Department of Health and Human Services.

## Additional information

### Competing interests

Ashish Lal: Reviewing editor, eLife. The other authors declare that no competing interests exist.

### Funding

| Funder | Grant reference number | Author |
|---|---|---|
| National Institutes of Health | ZIA BC011646 | Ashish Lal |

The funders had no role in study design, data collection and interpretation, or the decision to submit the work for publication.

### Author contributions

Ravi Kumar, Conceptualization, Data curation, Formal analysis, Validation, Methodology, Writing – original draft, Writing – review and editing; Simon Couly, Bruna R Muys, Mary Guest, Lisa M Jenkins, Raj Chari, Tsung-Ping Su, Methodology; Xiao Ling Li, Validation; Ioannis Grammatikakis, Data curation, Formal analysis, Writing – review and editing; Ragini Singh, Stefan Ambs, Investigation; Xinyu Wen, Data curation; Wei Tang, Erica C Pehrsson, Data curation, Formal analysis; Ashish Lal, Conceptualization, Resources, Formal analysis, Supervision, Investigation, Writing – original draft, Project administration, Writing – review and editing

### Author ORCIDs

Ioannis Grammatikakis ⓘ https://orcid.org/0000-0002-8455-1584
Lisa M Jenkins ⓘ https://orcid.org/0000-0003-1245-1338
Ashish Lal ⓘ https://orcid.org/0000-0002-4299-8177

Reviewer #1 (Public review): https://doi.org/10.7554/eLife.107538.3.sa1
Reviewer #2 (Public review): https://doi.org/10.7554/eLife.107538.3.sa2
Author response https://doi.org/10.7554/eLife.107538.3.sa3

# Additional files

### Supplementary files

MDAR checklist

Supplementary file 1. RNA-seq was performed from ZMAT3-WT and isogenic ZMAT3-KO HCT116 cells.

Supplementary file 2. Quantitative mass spectrometry was performed from ZMAT3-WT and ZMAT3-KO HCT116 cells.

Supplementary file 3. ZMAT3-regulated genes were identified by analyzing our previously published RNA-seq data from siCTRL and siZMAT3 transfected HCT116 cells in presence and absence of Nutlin-3 treatment (*Muys et al., 2021*).

Supplementary file 4. RNA-seq was performed from siCTRL and sip53 transfected HCT116 cells.

Supplementary file 5. Quantitative mass spectrometry was performed from ZMAT3-FLAG-HA IPs from doxycycline inducible ZMAT3-FLAG-HA HCT116 cells in presence and absence of doxycycline treatment.

Supplementary file 6. Sequences of oligos used for RT-qPCR, CRISPR-Cas9 knockout, ChIP-qPCR and HKDC1 intron 1 peak sequence used for reporter assays.

### Data availability

The RNA-seq data from ZMAT3-WT vs ZMAT3-KO HCT116 cells and from siCTRL vs sip53 from HCT116 have been deposited to GEO (https://www.ncbi.nlm.nih.gov/geo/query/acc.cgi?acc=GSE280756). The accession number is GSE280756.

The following dataset was generated:

| Author(s) | Year | Dataset title | Dataset URL | Database and Identifier |
|-----------|------|---------------|-------------|-------------------------|
| Lal A, Wen X | 2025 | Identification of RNAs regulated by ZMAT3 and p53 | https://www.ncbi.nlm.nih.gov/geo/query/acc.cgi?acc=GSE280756 | NCBI Gene Expression Omnibus, GSE280756 |

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
