## [Editor Report · eLife Assessment]

In this study, the authors investigate the role of ZMAT3, a p53 target gene, in tumor suppression and RNA splicing regulation. Using quantitative proteomics, the authors uncover that ZMAT3 knockout leads to upregulation of HKDC1, a gene linked to mitochondrial respiration, and that ZMAT3 suppresses HKDC1 expression by inhibiting c-JUN-mediated transcription. This set of **convincing** evidence reveals a **fundamental** mechanism by which ZMAT3 contributes to p53-driven tumor suppression by regulating mitochondrial respiration.

---

## [Referee Report · Reviewer #1 (Public review)]

Summary:

ZMAT3 is a p53 target gene that the Lal group and others have shown is important for p53-mediated tumor suppression, and which plays a role in the control of RNA splicing. In this manuscript Lal and colleagues perform quantitative proteomics of cells with ZMAT3 knockout and show that the enzyme hexokinase HKDC1 is the most upregulated protein. Mechanistically, the authors show that ZMAT3 does not appear to directly regulate the expression of HKDC1; rather, they show that the transcription factor c-JUN was strongly enriched in ZMAT3 pull-downs in IP-mass spec experiments, and they perform IP-western to demonstrate an interaction between c-JUN and ZMAT3. Importantly, the authors demonstrate, using ChIP-qPCR, that JUN is present at the HKDC1 gene (intron 1) in ZMAT3 WT cells, and showed markedly enhanced binding in ZMAT3 KO cells. The data best fit a model whereby p53 transactivates ZMAT3, leading to decreased JUN binding to the HKDC1 promoter, and altered mitochondrial respiration. The data are novel, compelling and very interesting.

Comments on revisions:

The authors have done a thorough job addressing my comments. This manuscript is quite strong and will be highly cited for its novelty and rigor.

---

## [Referee Report · Reviewer #2 (Public review)]

Summary:

The study elucidates the role of the recently discovered mediator of p53 tumor suppressive activity, ZMAT3. Specifically, the authors find that ZMAT3 negatively regulates HKDC1, a gene involved in the control of mitochondrial respiration and cell proliferation.

Comments on revisions:

The authors have mostly addressed to the concerns raised previously by this reviewer. The lack of functional assays made the reported findings mostly mechanistic with no clear biological context.

The present manuscript is certainly improved compared to the previous version.

---

## [Author Response]

The following is the authors’ response to the original reviews.

**Public Reviews:**

**Reviewer #1 (Public review):**
Summary:ZMAT3 is a p53 target gene that the Lal group and others have shown is important for p53mediated tumor suppression, and which plays a role in the control of RNA splicing. In this manuscript, Lal and colleagues perform quantitative proteomics of cells with ZMAT3 knockout and show that the enzyme hexokinase HKDC1 is the most upregulated protein. Mechanistically, the authors show that ZMAT3 does not appear to directly regulate the expression of HKDC1; rather, they show that the transcription factor c-JUN was strongly enriched in ZMAT3 pull-downs in IP-mass spec experiments, and they perform IP-western to demonstrate an interaction between c-JUN and ZMAT3. Importantly, the authors demonstrate, using ChIP-qPCR, that JUN is present at the HKDC1 gene (intron 1) in ZMAT3 WT cells and shows markedly enhanced binding in ZMAT3 KO cells. The data best fit a model whereby p53 transactivates ZMAT3, leading to decreased JUN binding to the HKDC1 promoter, and altered mitochondrial respiration.Strengths:The authors use multiple orthogonal approaches to test the majority of their findings. The authors offer a potentially new activity of ZMAT3 in tumor suppression by p53: the control of mitochondrial respiration.Weaknesses:Some indication as to whether other c-JUN target genes are also regulated by ZMAT3 would improve the broad relevance of the authors' findings.

We thank the reviewer for the kind words and the thoughtful suggestion. As recommended, to identify additional c-JUN targets potentially regulated by ZMAT3, we intersected the genes upregulated upon ZMAT3 knockout (from our RNA-seq data) with the ChIP-Atlas dataset for human c-JUN and cross-referenced these with c-JUN peaks from three ENCODE cell lines. From this analysis, we selected for further analysis the top 4 candidate genes - LAMA2, VSNL1, SAMD3, and IL6R (Figure 5-figure supplement 2A-D). Like HKDC1, these genes were upregulated in ZMAT3-KO cells, and this upregulation was abolished upon siRNA-mediated JUN knockdown in ZMAT3-KO cells (Figure 5-figure supplement 2E). Moreover, by ChIP-qPCR we observed increased JUN binding to the JUN peak for these genes in ZMAT3-KO cells as compared to the ZMAT3-WT (Figure 5- figure supplement 2F). As described on page 11 of the revised manuscript, these results suggest that the ZMAT3/JUN axis negatively regulates HKDC1 expression and additional c-JUN target genes.

**Reviewer #2 (Public review):**
Summary:The study elucidates the role of the recently discovered mediator of p53 tumor suppressive activity, ZMAT3. Specifically, the authors find that ZMAT3 negatively regulates HKDC1, a gene involved in the control of mitochondrial respiration and cell proliferation.Strengths:Mechanistically, ZMAT3 suppresses HKDC1 transcription by sequestering JUN and preventing its binding to the HKDC1 promoter, resulting in reduced HKDC1 expression. Conversely, p53 mutation leads to ZMAT3 downregulation and HKDC1 overexpression, thereby promoting increased mitochondrial respiration and proliferation. This mechanism is novel; however, the authors should address several points.Weaknesses:The authors conduct mechanistic experiments (e.g., transcript and protein quantification, luciferase assays) to demonstrate regulatory interactions between p53, ZMAT3, JUN, and HKDC1. These findings should be supported with functional assays, such as proliferation, apoptosis, or mitochondrial respiration analyses.

We thank the reviewer for appreciating our work and for this valuable suggestion. The reviewer rightly pointed out that supporting the regulatory interactions between p53, ZMAT3, JUN and HKDC1 with functional assays such as proliferation, apoptosis and mitochondrial respiration analyses would strengthen our mechanistic data. During the revision of our manuscript, we attempted to address this point by performing simultaneously knockdown of these proteins; however, we observed substantial toxicity under these conditions, making the functional assays technically unfeasible. This outcome was not unexpected as knockdown of JUN or HKDC1 individually results in growth defects. We therefore focused our efforts on addressing the recommendation for authors.

**Reviewer #3 (Public review):**
Summary:In their manuscript, Kumar et al. investigate the mechanisms underlying the tumor suppressive function of the RNA binding protein ZMAT3, a previously described tumor suppressor in the p53 pathway. To this end, they use RNA-sequencing and proteomics to characterize changes in ZMAT3-deficient cells, leading them to identify the hexokinase HKDC1 as upregulated with ZMAT3 deficiency first in colorectal cancer cells, then in other cell types of both mouse and human origin. This increase in HKDC1 is associated with increased mitochondrial respiration. As ZMAT3 has been reported as an RNA-binding and DNA-binding protein, the authors investigated this via PAR-CLIP and ChIP-seq but did not observe ZMAT3 binding to HKDC1 pre-mRNA or DNA. Thus, to better understand how ZMAT3 regulates HKDC1, the authors used quantitative proteomics to identify ZMAT3interacting proteins. They identified the transcription factor JUN as a ZMAT3-interacting protein and showed that JUN promotes the increased HKDC1 RNA expression seen with ZMAT3 inactivation. They propose that ZMAT3 inhibits JUN-mediated transcriptional induction of HKDC1 as a mechanism of tumor suppression. This work uncovers novel aspects of the p53 tumor suppressor pathway.Strengths:This novel work sheds light on one of the most well-established yet understudied p53 target genes, ZMAT3, and how it contributes to p53's tumor suppressive functions. Overall, this story establishes a p53-ZMAT3-HKDC1 tumor suppressive axis, which has been strongly substantiated using a variety of orthogonal approaches, in different cell lines and with different data sets.Weaknesses:While the role of p53 and ZMAT3 in repressing HKDC1 is well substantiated, there is a gap in understanding how ZMAT3 acts to repress JUN-driven activation of the HKDC1 locus. How does ZMAT3 inhibit JUN binding to HKDC1? Can targeted ChIP experiments or RIP experiments be used to make a more definitive model? Can ZMAT3 mutants help to understand the mechanisms? Future work can further establish the mechanisms underlying how ZMAT3 represses JUN activity.

We thank the reviewer for the kind words and the invaluable suggestion. The reviewer has an excellent point regarding how ZMAT3 inhibits JUN binding to HKDC1 locus.Our new data included in the revised manuscript show that the ZMAT3-JUN interaction is lost in the presence of DNase or RNase, indicating that the interaction requires both DNA and RNA. This result suggests that ZMAT3 and JUN form an RNA-dependent, chromatin- associated complex. Although not directly investigated in our study, this finding is consistent with emerging evidence that RBPs can function as chromatin-associated cofactors in transcription. For example, functional interplay between transcription factor YY1 and the RNA binding protein RBM25 co-regulates a broad set of genes, where RBM25 appears to engage promoters first and then recruit YY1, with RNA proposed to guide target recognition. We have discussed this possibility in the discussion section of revised manuscript (page 13). We agree that future work using ZMAT3 mutants and targeted ChIP or RIP assays will be valuable to delineate the precise mechanism by which ZMAT3 inhibits JUN binding to its target genes.

**Recommendations for the authors:**

**Reviewer #1 (Recommendations for the authors):**
ZMAT3 is a p53 target gene that the Lal group and others have shown is important for p53mediated tumor suppression, and which plays a role in the control of RNA splicing. In this manuscript, Lal and colleagues perform quantitative proteomics of cells with ZMAT3 knockout and show that the enzyme hexokinase HKDC1 is the most upregulated protein. HKDC1 is emerging as an important player in human cancer. Importantly, the authors show both acute (gene silencing) and chronic (CRISPR KO) approaches to silence ZMAT3, and they do this in several cell lines. Notably, they show that ZMAT3 silencing leads to impaired mitochondrial respiration, in a manner that is rescued by silencing of HKDC1. Mechanistically, the authors show that ZMAT3 does not appear to directly regulate the expression of HKDC1; rather, they show that the transcription factor c-JUN was strongly enriched in ZMAT3 pull-downs in IP-mass spec experiments, and they perform IP-western to demonstrate an interaction between c-JUN and ZMAT3. Importantly, the authors demonstrate, using ChIP-qPCR, that JUN is present at the HKDC1 gene (intron 1) in ZMAT3 WT cells, and shows markedly enhanced binding in ZMAT3 KO cells. The data best fit a model whereby p53 transactivates ZMAT3, leading to decreased JUN binding to the HKDC1 promoter (intron 1), and altered mitochondrial respiration. The findings are compelling, and the authors use multiple orthogonal approaches to test most findings. And the authors offer a potentially new activity of ZMAT3 in tumor suppression by p53: the control of mitochondrial respiration. As such, enthusiasm is high for this manuscript.Addressing the following question would improve the manuscript.It is not clear how many (other) c-JUN target genes might be impacted by ZMAT3; other important c-JUN targets in cancer include GLS1, WEE1, SREBP1, GLUT1, and CD36, so there could be a global impact on metabolism in ZMAT3 KO cells. Can the authors perform qPCR on these targets in ZMAT3 WT and KO cells and see if these target genes are differentially expressed?

We thank the reviewer for this thoughtful suggestion. As recommended, we examined the expression of key c-JUN target genes GLS1 (also known as GLS), WEE1, SREBP1, GLUT1, and CD36 in ZMAT3-WT and ZMAT3-KO cells. We first analyzed publicly available JUN ChIP-Seq data from three ENCODE cell lines, which revealed JUN binding peaks near or upstream of exon 1 for GLS1/GLS, SREBP1, and SLC2A1/GLUT1, but not for WEE1 or CD36 (Appendix 1, panels A-E). Based on these results, we performed RT-qPCR for GLS1/GLS, SREBP1 and SLC2A1 in ZMAT3-WT and ZMAT3-KO cells, with or without JUN knockdown. GLS mRNA was significantly reduced upon JUN knockdown in both ZMAT3-WT cells and ZMAT3-KO cells, but it was not upregulated upon loss of ZMAT3, indicating that GLS is a JUN target gene, but it is not regulated by ZMAT3. In contrast, SREBF1 or SLC2A1 expression remained unchanged upon ZMAT3 loss or JUN knockdown (Appendix 1 panels F-H). These data suggest that the ZMAT3/JUN axis does not regulate the expression of these genes.

To identify additional c-JUN targets potentially regulated by ZMAT3, we intersected the genes upregulated upon ZMAT3 knockout (from our RNA-seq data) with the ChIP-Atlas dataset for human c-JUN and cross-referenced these with c-JUN peaks from three ENCODE cell lines. From this analysis, we selected for further analysis the top 4 candidate genes - LAMA2, VSNL1, SAMD3, and IL6R (Figure 5-figure supplement 2A-D). Like HKDC1, these genes were upregulated in ZMAT3-KO cells, and this upregulation was abolished upon siRNA-mediated JUN knockdown in ZMAT3-KO cells (Figure 5-figure supplement 2E). Moreover, by ChIP-qPCR we observed increased JUN binding to the JUN peak for these genes in ZMAT3-KO cells as compared to the ZMAT3-WT (Figure 5- figure supplement 2F). As described on page 11 of the revised manuscript, these results suggest that the ZMAT3/JUN axis negatively regulates HKDC1 expression and additional c-JUN target genes.

Minor concerns:(1) Line 150: observed a modest.(2) Line 159: Figure 2G appears to be inaccurately cited.(3) Line 191: assays to measure.

We thank the reviewer for pointing these out. These minor concerns have been addressed in the text.

**Reviewer #2 (Recommendations for the authors):**
(1) Figure 1E: Can the authors clarify what the numbers on the left side of the chart represent? Do they refer to the scale?

The numbers on the Y-axis represent the -log 10 (p- value) where higher values correspond to more significant changes. For visualization purposes, the significant changes are shown in red.

(2) Page 5, line 123: The sentence "As expected, ZMAT3 mRNA levels were decreased in the ZMAT3-KO cells" is redundant, as this information was already mentioned on page 4, line 103.

We thank the reviewer for noticing this redundancy. The repeated sentence has been removed in the revised manuscript.

(3) Page 5: The authors state: "Transcriptome-wide, upon loss of ZMAT3, 606 genes were significantly up-regulated (adj. p < 0.05 and 1.5-fold change) and 552 were down-regulated, with a median fold change of 1.76 and 0.55 for the up- and down-regulated genes, respectively." Later, on page 6, they write: "Comparison of the RNA-seq data from ZMAT3WT vs. ZMAT3-KO and CTRL siRNA vs. ZMAT3 siRNA-transfected HCT116 cells indicated that 1023 genes were commonly up-regulated, and 1042 were commonly down-regulated upon ZMAT3 loss (Figure S2C and D)." Why is the number of deregulated transcripts higher in the ZMAT3-WT vs. ZMAT3-KO comparison than in the CTRL siRNA vs. ZMAT3 siRNA comparison? Are the authors using less stringent criteria in the second analysis? This point should be clarified.

We thank the reviewer for highlighting this point. The reviewer is correct that less stringent criteria were used in the second analysis. On page 5, we applied stringent thresholds (adjusted p-value < 0.05 and 1.5-fold change) to identify high-confidence transcriptome-wide changes upon ZMAT3 loss. In contrast, for the comparison of both RNA-seq datasets (ZMAT3-WT vs. KO and siCTRL vs. siZMAT3), we included genes that were consistently up- or downregulated, without applying a fold change threshold, focusing instead on significantly altered genes (adjusted p < 0.05) in both datasets. This allowed us to capture broader and more reproducible transcriptomic changes that occur upon ZMAT3 depletion, including modest but significant changes upon transient ZMAT3 knockdown with siRNAs. We have now clarified this distinction on page 6 of the revised manuscript.

(4) Figures 2B and 2E: The authors should provide quantification of HKDC1 protein levels normalized to a loading control. In addition, they should assess HKDC1 protein abundance upon ZMAT3 interference in SWI1222 and HCEC1CT cells, not just in HepG2 and HCT116 cells.

We thank the reviewer for this suggestion. We have now quantified all immunoblots presented throughout the manuscript, including those shown in Figures 2B and 2E, and all other figures containing protein analyses. Band intensities were quantified using ImageJ densitometry and normalized to GAPDH as the loading control. In addition, as suggested, we examined HKDC1 protein levels following ZMAT3 knockdown in two additional cell lines, SW1222 and HCEC-1CT. Consistent with our observations in HepG2 and HCT116 cells, ZMAT3 depletion led to increased HKDC1 protein levels in both SW1222 and HCEC-1CT cells. These new data are now included in Figure 2-figure supplement 1F and G. We have updated the Results section, figure legends, and figures to reflect these additions.

(5) Figure 3A: It is unclear which gene was knocked out in the "KO cells." The authors should clearly specify this.

We thank the reviewer for pointing this out. We have now updated Figure 3A.

(6) Figure 3D: The result appears counterintuitive in comparison to Figure 3E. Why does HKDC1 knockdown reduce cell confluency more in ZMAT3 KO cells than in control (ZMAT3 wild-type) cells? The authors should explain this discrepancy more clearly.

We thank the reviewer for this insightful comment. As shown in Figure 3D and 3E, knockdown of HKDC1 resulted in a greater decrease in proliferation in ZMAT3-KO cells than in ZMAT3-WT cells. This observation was indeed unexpected, given that HKDC1 acts downstream of ZMAT3. One possible explanation is that elevated HKDC1 expression in ZMAT3-KO cells increases their reliance on HKDC1 for sustaining proliferation, and that HKDC1 may also participate in additional pathways in ZMAT3-KO cells. Consequently, transient knockdown of HKDC1 in ZMAT3-KO cells would have a more pronounced effect on proliferation due to their increased dependency on HKDC1 activity. In contrast, ZMAT3WT cells which express lower levels of HKDC1 are less dependent on its function and therefore less sensitive to its depletion. We have now clarified this point on page 8 of the revised manuscript.

**Reviewer #3 (Recommendations for the authors):**
(1) Why do the authors start their analysis by knocking out the p53 response element in Zmat3? That should be clarified. In addition, since clones were picked after CRISPR KO of Zmat3, were experiments done to confirm that p53 signaling was not disrupted?

We thank the reviewer for this thoughtful question. We began our study by targeting the p53 response element (p53RE) in the ZMAT3 locus because the basal expression of ZMAT3 is regulated by p53 (Muys, Bruna R. et al., Genes & Development, 2021). Deleting the p53RE therefore allowed us to markedly reduce ZMAT3 expression without disrupting the entire ZMAT3 locus. We have clarified this rationale on page 4 of the revised manuscript. To ensure that p53 signaling was not affected by this modification, we verified that canonical p53 targets such as p21 were equivalently induced in both ZMAT3WT and KO cells following Nutlin treatment and that p53 induction was unchanged(Figure 4F and Figure 1 – figure supplement 1A).

(2) Throughout the text, many immunoblots are used to validate the knockouts and knockdowns used, but some clarification is needed. In Figure S1A, the Zmat3-WT sample seems to have significantly more p53 than the Zmat3 KO sample. Does Zmat3 KO compromise p53 levels in other experiments? It would be good to understand if Zmat3 affects p53 function by affecting its levels. Also, the p21 blot is overloaded.

We thank the reviewer for this helpful observation. To determine whether ZMAT3 knockout affects p53 function by affecting its levels, we repeated the experiment three independent times. Western blots from these biological replicates, together with protein quantification, are now included in Appendix-2 and Figure 1-figure supplement 1A. These data show no significant differences in p53 or p21 induction between ZMAT3-WT and ZMAT3-KO cells following Nutlin treatment. In the revised manuscript, we have replaced the blot in Figure 1-figure supplement 1A with a more representative image from one of these replicate experiments.

In Figure 2E, HKDC1 protein levels are not shown for the SW1222 and HCEC-1CT cell lines,

We thank the reviewer for this suggestion. HKDC1 protein levels in SW1222 and HCEC1-CT cells following ZMAT3 knockdown are now included as Figure 2- figure supplement 1F and 1G, together with the corresponding quantification.

and Zmat3 does not appear as its characteristic two bands on the blot. What does this signify?

We thank the reviewer for this observation. Endogenous ZMAT3 typically appears as two closely migrating bands on immunoblots. As shown in Figure 4D and Appendix 2A and 2B, these two bands are observed at the expected molecular weight following Nutlin treatment and are specific to ZMAT3, as they are markedly reduced in ZMAT3-KO cells. In contrast, only a single ZMAT3 band is visible in Figure 2E. This likely reflects limited resolution of the two bands in some blots rather than a biological difference.

(3) Why does HKDC1 knockdown only have an effect on metabolic phenotypes when ZMAT3 is gone? In Figure 3A, there does not seem to be a decrease in hexokinase activity in the siCTRL + siHKDC1 condition compared to siCTRL alone. Also, in Figure 3A, does phosphorylation activity of HKDC1 necessarily reflect glucose uptake, as stated? Additionally, in Figure 3C, there is no effect on mitochondrial respiration with siHKDC1, even though recent studies have shown a significant effect of HKDC1 on this.

We thank the reviewer for raising these important questions. As noted, HKDC1 knockdown alone in wild-type cells (siCTRL + siHKDC1) does not significantly reduce hexokinase activity (Figure 3A). This likely reflects the low basal expression of HKDC1 in these cells. Thus, the metabolic phenotype may only become apparent when HKDC1 expression exceeds a functional threshold, as observed in ZMAT3-KO cells where HKDC1 is upregulated.

Regarding the glucose uptake assay, HKDC1 itself is not phosphorylated; rather, it phosphorylates a non-catabolizable glucose analog, 2-deoxyglucose (2-DG) upon cellular uptake. According to the manufacturer’s protocol, intracellular 2-DG is phosphorylated by hexokinases to 2-deoxyglucose-6-phosphate (2-DG6P), which cannot be further metabolized and therefore accumulates. The accumulated 2-DG6P is quantified using a luminescence-based readout. This assay is widely used as a surrogate for glucose uptake because it reflects both glucose import and phosphorylation — the first step of glycolytic flux. As for the lack of change in mitochondrial respiration (Figure 3C), we acknowledge that some studies have reported mitochondrial roles for HKDC1 under basal conditions; however, such effects may be cell type-specific.

(4) The emphasis on glycolysis signatures is confusing, as in the end, glycolysis does not seem to be affected by ZMAT3 status, but mitochondrial respiration is affected. Can the text be clarified to address this? It is also difficult to understand the role of oxygen consumption rate (OCR) in ZMAT3 phenotypes, as it does not fully track with proliferation. For example, ZMAT3 KD has the highest OCR, and the other conditions have similar OCRs but different proliferative rates in Figure 3D. Also, the colors used in Figure 3 to denote different genotypes change between B/C and D, which is confusing.

We thank the reviewer for pointing out the inconsistency in the colors of the graph in Figure 2, which we have now corrected. Our data indicates that ZMAT3 regulates mitochondrial respiration without significantly affecting glycolysis. It is possible that mitochondria in ZMAT3-KO cells are oxidizing more substrates that are not produced by glycolysis. Additional work will be required to fully determine these mechanisms. We have clarified this on page 8 of the revised manuscript.

(5) The lack of ZMAT3 binding to RNAs in PAR-CLIP is not proof that it does not do so. A more targeted approach should be used, using individual RIP assays. The authors should also analyze the splicing of HKDC1, which could be affected by ZMAT3.

As suggested, we performed ZMAT3 RNA IP experiments (RIP) using doxycycline-inducible HCT116-ZMAT3-FLAG cells. However, we did not observe significant enrichment of HKDC1 mRNA in the ZMAT3 IPs (Figure 5 – figure supplement 1A), consistent with previously published ZMAT3 RIP-seq data (Bersani et al, Oncotarget, 2016). These findings further support the notion that ZMAT3 does not directly bind to HKDC1 mRNA in these cells. We Accordingly, we have modified the text on page 10 of the revised manuscript.

In addition, as suggested by the reviewer, we analyzed changes in splicing of HKDC1 pre-mRNA using rMATS in HCT116 cells by comparing our previously published RNA-seq data from siCTRL and siZMAT3-transfected HCT116 cells (Muys et al, Genes Dev, 2021). We focused on splicing events with an FDR < 0.05 and a delta PSI > |0.1| (representing at least a 10% change in splicing). The splicing analysis (data not shown) did not reveal any significant alterations in HKDC1 pre-mRNA splicing upon ZMAT3 knockdown. Corresponding text has been updated on page 10 of the revised manuscript.

(6) The authors say that they examine JUN binding at the HKDC1 promoter several times, but they focus on intron 1 in Figure 5. They should revise the text accordingly, and they should also show JUN ChIP data traces for the whole HKDC1 locus in Figure 5C.

We thank the reviewer for this helpful suggestion. As recommended, we have revised the text throughout the manuscript and replaced HKDC1 promoter with HKDC1 intron 1 DNA to accurately reflect our analysis, and Figure 5 now shows the JUN ChIP-seq signal across the entire HKDC1 locus.

(7) In the ZMAT3 and JUN interaction assays, were these tested in the presence of DNAse or RNAse to determine if nucleic acids mediate the interaction?

We thank the reviewer for this valuable suggestion. To test whether nucleic acids mediate the ZMAT3-JUN interaction, we performed ZMAT3 immunoprecipitation (IPs) in the presence or absence of DNase and RNase from doxycycline-inducible ZMAT3-FLAG expressing HCT116 cells. The ZMAT3-JUN interaction was lost upon treatment with either DNase or RNase, indicating that the interaction is mediated by nucleic acids. This data has been added in the revised manuscript (Figure 5-figure supplement 1D and on page 11).